# Only Large Weights (And Not Skip Connections) Can Prevent the Perils of Rank Collapse

## Abstract

Attention mechanisms lie at the heart of modern large language models (LLMs). Straightforward algorithms for forward and backward (gradient) computation take quadratic time, and a line of work initiated by [Alman and Song NeurIPS 2023] and [Alman and Song NeurIPS 2024] has shown that quadratic time is necessary unless the model weights are small, in which case almost linear time algorithms are possible. In this paper, we show that large weights are necessary to avoid a strong preclusion to representational strength we call layer collapse, which means that the entire network can be approximated well by a network with only a single layer. This means that transformers with small weights are shockingly weak, and that the quadratic running time of attention is unavoidable for expressive transformers.

The notion of layer collapse that we introduce is a variant on the notion of rank collapse from the work of [Dong, Cordonnier, and Loukas ICML 2021]. They showed that in Self Attention Networks with small weights and with skip connections, rank collapse must occur. This is typically interpreted as justifying the necessity of skip connections in expressive networks. However, our result shows that even with skip connections, if the weights are small, then layer collapse still occurs. Thus, only large weights, and not skip connections, can prevent these representational weaknesses.

## 1 Introduction

The rapid progress of large language models, text-to-image and text-to-video models like Transformer (Vaswani et al., 2017), BERT (Devlin et al., 2018), GPT-4 (OpenAI, 2023), Llama 3 (Llama Team, 2024), and Gemini 2.0 (Google, 2025), has enabled powerful language modelling abilities. These models take advantage of large-scale pretraining on massive textual data, which equips them with strong abilities to interpret the complex patterns of natural language. These LLMs have a broad range of applications, influencing domains such as human-computer interaction, multilingual translation, language comprehension, text generation, and rapid prototyping of software.

The major architecture behind the success of all these language models is the attention mechanism. Specifically, attention computes pairwise similarities by calculating inner products between vectorized representations of words, with input sequences represented as vectors. Formally, softmax attention can be formulated as follows:

**Definition 1.1** (Self-Attention with Softmax Units). *Let $A \in \mathbb{R}^{n \times d}$ and weights $Q, K, V \in \mathbb{R}^{d \times d}$. Let $g$ represent the entry-wise exponentiation function, i.e., for $z \in \mathbb{R}$ we have $g(z) = \exp(z)$, and for a matrix $W$ we have $g(W)_{i,j} = g(W_{i,j})$. The attention computation can be defined as*

$$\mathsf{SAtt}(X, Q, K, V) = \underbrace{D^{-1}}_{n \times n} \underbrace{g(XQK^\top X^\top)}_{n \times n} \underbrace{X}_{n \times d} \underbrace{V}_{d \times d}$$

*where $D := \mathrm{diag}(g(XQK^\top X^\top)\mathbf{1}_n)$, and where $\mathbf{1}_n \in \mathbb{R}^n$ is a length-$n$ vector whose entries are all 1.*

**Small Coefficients are Needed for Fast Algorithms** However, the straightforward algorithm for computing self-attention results in a quadratic $O(n^2 d)$ running time, where $n$ is the length of the input token and $d$ is the hidden dimension. Under popular complexity-theoretic assumptions, there is no better, subquadratic time algorithm to compute attention, even approximately (Alman & Song, 2023). Therefore, models based on attention may face difficulties when they handle long contexts.

In fact, a key observation of this line of work on the computational complexity of attention is that attention can be computed (or tightly approximated) faster if one restricts to small weights, i.e., an upper bound on how large the entries of $Q, K, V$ can be in Definition 1.1 above. Indeed, a line of work (Alman & Song, 2023; 2024a;b; 2025) has shown that small weights are both necessary and sufficient for a faster algorithm: If the weights are large, then the aforementioned complexity-theoretic result shows that there is no subquadratic time algorithm. However, if the weights are small, then attention can be approximated to low error in *almost linear time*! Their algorithm is based on low-rank approximations of the $n \times n$ attention matrix (the matrix $g(XQK^\top X^\top)$ in Definition 1.1 above).

This type of observation is also frequently used in practice; many LLM implementations have enforced bounds on the weights, often using techniques like approximation or quantization, and then used this for substantial speedups. For some examples, see (Zafrir et al., 2019; Katharopoulos et al., 2020b; Frantar et al., 2022; Perez et al., 2023; Dettmers et al., 2023; Egashira et al., 2024; Liu et al., 2024b; Xu et al., 2024a; Lin et al., 2025; Chen et al., 2025b; Liu et al., 2025; Ouyang et al., 2025; Deng et al., 2025; Hu et al., 2025c; Fu et al., 2025; Hu et al., 2025b; Park et al., 2025; Zeng et al., 2025; Yu et al., 2025; Wei et al., 2025).

In this paper, we investigate the representational strength of transformers with small weights. Our main result will show a limitation, that without large weights, a transformer cannot take advantage of more than a single layer. In other words, we will show that in order to take advantage of the full expressive power of the transformer model, large weights are necessary.

**Rank Collapse and Skip Connections** We will crucially build on the approach of Dong, Cordonnier, and Loukas (Dong et al., 2021), who studied the representational strength of different variants on the transformer architecture through the lens of a notion called *rank collapse*. We say that a model experiences rank collapse if, on any input, the output must always be close to a rank 1 matrix. (See Definition 3.4 below for the precise meaning.) Beyond being unable to represent complex concepts, models with rank collapse also have numerous other issues in both training and evaluation (Noci et al., 2022; Roth & Liebig, 2024; Naderi et al., 2024; Nguyen et al., 2024; Heo & Choi, 2024; Yuan & Xu, 2024; Barbero et al., 2025; Bonino et al., 2025).

The work of (Dong et al., 2021) highlights *skip connections* (or residual connections) in a transformer network as crucial for avoiding rank collapse. They show that in a Self-Attention Network without skip connections, rank collapse occurs with a doubly exponential rate of convergence. More precisely, if $\beta$ is a bound on the $\ell_1$ norm of the weight matrices of the network, and the network has $L$ layers, then they show the distance to a rank-1 matrix shrinks as

$$O(\beta)^{\frac{3^L-1}{2}}. \tag{1}$$

Meanwhile, they observe that networks with skip connections may experience no rank collapse at all. For instance, it is not hard to simulate the *identity* function as a Self-Attention Network with skip connections (simply set all value weights to 0, so that only the skip connections are output). In this case, any input which is far from rank-1 will result in an output which is also far from rank-1. They study other mechanisms in transformer networks as well, including multi-layer perceptrons and layer normalization, but find that only skip connections prevents the rank collapse of Equation (1). This result is frequently cited in the literature as evidence of the importance of skip connections (Ma et al., 2021; Noci et al., 2022; Sander et al., 2022; Guo et al., 2023; Li et al., 2023; Kim et al., 2023; Geshkovski et al., 2023; Kim et al., 2024; Ji et al., 2025).

**The Importance of Large Weights and Layer Collapse** We begin with a simple observation: in order for Equation (1) to be shrinking as $L$ grows, it is necessary that $\beta$ is small, i.e., that the weights of the network are small. In other words, the result of (Dong et al., 2021) really says that:

> To avoid rank collapse, one needs either skip connections *or* large weights.

In this paper, we prove that Self Attention Networks with skip connections, but with small weights, must suffer from a phenomenon similar to rank collapse which we call *layer collapse*. We say that an $L$-layer Self Attention Network $S$ has layer collapse if there is a nearly equivalent Self Attention Network $S'$ which only has a single layer. In other words, although $S'$ only has one layer, it is still as expressive as $S$, since on any input $X$, the outputs $S(X)$ and $S'(X)$ differ in each entry by at most a small error parameter.

When combined with (Dong et al., 2021), our result implies:

> To avoid rank and layer collapse, one needs large weights (skip connections do not suffice).

This challenges the previous popular interpretation of (Dong et al., 2021), that skip connections were crucial for the representational strength of the model.

The connection between layer collapse and rank collapse may not be evident from the definitions, but it will become clear in our proofs below. At a high level, we will find that the attention mechanisms in lower layers of the Self Attention Network must exhibit rank collapse (regardless of skip connections), and can thus be removed from the network without substantially changing the output. We will show

**Theorem 1.2** (Main result, informal). *If $S$ is a Self Attention Network whose weight matrices have $\ell_\infty$ norm bounded by $\eta$, then there is a Self Attention Network $S'$ with only one layer, such that on any input $X$ with $\|X\|_\infty \leq O(1)$, we have $\|S(X) - S'(X)\|_\infty \leq O(\eta)$.*

In fact, the example from (Dong et al., 2021) of the identity network with skip connections heavily inspired our definition of layer collapse. That network indeed does not have rank collapse, so we could not hope to prove a version of Theorem 1.2 with rank collapse instead of layer collapse. On the other hand, it is essentially not making use of its attention mechanisms; they could be removed without changing the output of the network. Our key idea is to show that, more generally, the attention mechanisms with small weights can be removed from any Self Attention Network, with skip connections, without changing the output of the network by very much.

Our $\eta$ in Theorem 1.2 is a bound on the $\ell_\infty$ norm of the weight matrices (maximum magnitude of an entry), whereas the prior result in Eq. (1) above uses parameter $\beta$, which is a bound on the $\ell_1$ norm (sum of magnitudes of all entries). Our $\eta$ could thus be quite a bit smaller (by a factor of $d^2$ for $d \times d$ weight matrices), and there are thus networks without skip connections where (Dong et al., 2021) does not imply rank collapse (since $\beta \gg 1$ is too big) but our Theorem 1.2 still implies layer collapse (since $\eta \ll 1$ is smaller).

We also note that both our informal statement of Theorem 1.2 and our presentation of the main result of (Dong et al., 2021) in Eq. (1), are given assuming that the Self Attention Network has a constant number of heads and layers. The more complete statement in terms of the number of heads and layers is presented in Theorem D.1 in the appendix. Both results have modest assumptions on the relationships between $\eta$ (or $\beta$), $\|X\|_\infty$, and the numbers of heads and layers, and we emphasize that these assumptions are nearly identical in both results; see Remark D.3 for more details.

**Roamdap.** In Section 2, we present the related work. In Section 3, we introduce several basic notations and definitions. In Section 4, we study perturbation properties of several functions, such as softmax. In Section 5, we provide several major rank collapse results. In Section 6, we provide the conclusion of this paper.

## 2 RELATED WORK

**Low-rank Approximations** Low rank approximation is a fundamental topic in numerical linear algebra (Clarkson & Woodruff, 2013; Nelson & Nguyên, 2013; Song et al., 2023b;a). Many problems require either computationally or analytically finding a low-rank approximation under different settings such as linear and kernel SVMs (Gu et al., 2025), tensor regression (Song et al., 2021b; Reddy et al., 2022; Diao et al., 2018; 2019), low rank approximation with Frobenious norm (Clarkson & Woodruff, 2013; Nelson & Nguyên, 2013), weighted low rank approximation (Razenshteyn et al., 2016; Gu et al., 2024; Li et al., 2025a; Song et al., 2025), general norm column subset selection (Song et al., 2019a), entrywise $\ell_1$ norm low rank approximation (Song et al., 2017; 2019b),

tensor low rank approximation (Song et al., 2019c), tensor power method (Deng et al., 2023b), and matrix CUR decomposition (Boutsidis & Woodruff, 2014; Song et al., 2017; 2019c). Rank collapse and other techniques we use here build on this line of work.

**Algorithmic Result for Attention Computations** The quadratic time complexity of attention mechanisms (Vaswani et al., 2017) has posed significant computational challenges for long sequences. In response to this problem, a wide range of works have been proposed to reduce computational cost and enhance the scalability of attention mechanisms, including sparsification (Child et al., 2019; Zaheer et al., 2020; Beltagy et al., 2020; Hubara et al., 2021; Shi et al., 2023a; Kurtic et al., 2023; Frantar & Alistarh, 2023; Li et al., 2024b; Liang et al., 2024a; Han et al., 2024), kernel-based approaches (Liu & Zenke, 2020; Charikar et al., 2020; Zandieh et al., 2023; Deng et al., 2023a; Liang et al., 2024b), and low-rank methods (Li et al., 2016; Razenshteyn et al., 2016; Hu et al., 2022; 2024b; Zeng & Lee, 2024). Additionally, another promising line of research is linear attention (Tsai et al., 2019; Katharopoulos et al., 2020a; Schlag et al., 2021; Deng et al., 2023c; Sun et al., 2023; Zhang et al., 2023b; Ahn et al., 2024; Li et al., 2024a; Shi et al., 2023c; Zhang et al., 2024), which significantly accelerates traditional softmax attention. Other relevant works have explored important aspects of attention mechanisms, covering topics such as circuit complexity (Chen et al., 2024a;c; Li et al., 2025b), model pruning (Frantar & Alistarh, 2023; Shen et al., 2024; Sun et al., 2024; Liang et al., 2025), privacy protection (Liang et al., 2024d; Gao et al., 2024), regression (Gao et al., 2023b), half-space reporting (HSR) (Jiang et al., 2021; Chen et al., 2024b), and quantum computation (Gao et al., 2023c; Zhao et al., 2024).

**Polynomial Kernels for Attention Acceleration** With the assumption that model weights are small, polynomial kernels (Alman & Song, 2023; 2024b) are powerful tools for approximating attention computation in almost linear time complexity, providing promising acceleration for both training and inference of a single attention layer. This approach can be further extended to a wide range of applications. For instance, polynomial kernels can provide insights into novel attention mechanisms and model designs, such as modern Hopfield models (Hu et al., 2024a), Diffusion Transformers (DiTs) (Shen et al., 2025; Hu et al., 2024d), multi-layer Transformers (Liang et al., 2024c), and tensor attention mechanisms (Liang et al., 2024e; Alman & Song, 2024a). These polynomial kernel methods also contribute to efficient and model-utility-preserving fine-tuning of foundation models, such as model adapters (Hu et al., 2022; Zhang et al., 2023a; Shi et al., 2023b; Gao et al., 2023a), multi-task fine-tuning (Gao et al., 2021; Oswald et al., 2023; Xu et al., 2024b), black-box model tuning (Sun et al., 2022), and instruction tuning (Li & Liang, 2021; Chung et al., 2022; Mishra et al., 2022). Other promising applications include privacy protection in attention computation (Liang et al., 2024d), CoT reasoning (Khattab et al., 2022; Wei et al., 2022; Yao et al., 2023; Zheng et al., 2024), and model calibration (Zhao et al., 2021; Zhou et al., 2023). Very recently, (Gupta et al., 2025) further extends the work of (Alman & Song, 2023) to almost all the regimes of parameter $d$ (see definition of $d$ in Defintion 1.1).

**Regression Models** The unprecedented energy consumption in training large-scale ML models has necessitated the development of scalable and efficient ML models (Venkataramani et al., 2015; Bender et al., 2021; McDonald et al., 2022). As a simple yet powerful approach to solving various machine learning problems (Bubeck, 2015; Brand et al., 2021; Song et al., 2024b; Subrahmanya & Shin, 2009), simple regression models have raised significant concerns in model acceleration, with recent advances from different perspectives, including sketching (Song & Yu, 2021; Reddy et al., 2022; Song et al., 2023a) and pre-conditioning (Yang et al., 2018; Kelner et al., 2022; Song et al., 2024a). Our work discusses low-rank approximations in attention mechanisms, while our general insight can be extended to other low-rank method applications, such as accelerated regression models.

**Diffusion Models** Diffusion models and score-based generative models have achieved remarkable success in generating human-preference-aligned and high-quality visual content (Ho et al., 2020; Song et al., 2021a; Blattmann et al., 2023). These advances not only benefit vision tasks but also enhance the performance of other applications, such as language modeling (Lin et al., 2023; Sahoo et al., 2024), chemical design (Xu et al., 2023; Wen et al., 2024), and e-commerce (Yang et al., 2023; Wang et al., 2023; Liu et al., 2024a). Relevant works have discussed the theoretical guarantee that diffusion models can be approximated efficiently (Hu et al., 2024d; 2025a; 2024c;

Gong et al., 2025). Empirical approaches to accelerate diffusion models have addressed various aspects, such as shortcuts (Frans et al., 2024; Dao et al., 2024; Chen et al., 2025a), parameter pruning (Castells et al., 2024; Ma et al., 2024), and lazy computation (Nitzan et al., 2024; Shen et al., 2025). With these acceleration techniques, diffusion models can be trained on larger-scale data, overcoming inherent limitations such as counting (Hui et al., 2024; Cao et al., 2025; Guo et al., 2025a), text rendering (Chen et al., 2023; Tuo et al., 2024; Guo et al., 2025c), and adherence to physical constraints (Motamed et al., 2025; Guo et al., 2025b; Bansal et al., 2025). Most diffusion models leverage Transformer backbones for enhanced modelling capability. Our work accelerates attention mechanism computations, significantly benefiting a wide range of diffusion models.

**Graph ML Models** Relational data is prevalent in many real-world scenarios, where graph neural networks (GNNs) are the powerful solutions for mining effective patterns from such relations (Kipf & Welling, 2017; Hamilton et al., 2017; Wu et al., 2019). Recent scalability approaches have widely adopted low-rank approximations, such as sketching (Ding et al., 2022; Chamberlain et al., 2023) and vector quantization (Ding et al., 2021; Wang et al., 2025), which can take insights from this paper. These accelerations empower a wide range of applications, including misleading information mitigation (Xu et al., 2022; Chang et al., 2024), social network prediction (Fan et al., 2019; Zhang et al., 2022), and human action recognition (Peng et al., 2020; Li et al., 2021; Fu et al., 2021), while also inspiring advances in multiple aspects of graph learning, such as differential privacy (Lin et al., 2022; Mueller et al., 2022), robustness (Geisler et al., 2021; Dai et al., 2022; Zeng et al., 2022), and sensitive data removal (Chien et al., 2023; Zhang, 2024; Yi & Wei, 2025). A recent work (Zhang, 2024) proposes an efficient framework for empowering sensitive data impact removal from trained GNNs with partial retraining, leveraging model utility-aware data partitioning and contrastive sub-model aggregation.

## 3 PRELIMINARIES

In Section 3.1, we provide basic notation, definitions and facts. In Section 3.2 and Section 3.3, we define the Res function and balanced matrix notation which will appear prominently in our constructions. In Section 3.4, we provide the definition of a multi-layer multi-head Self Attention Network which we study here.

### 3.1 BASIC NOTATION AND FACTS

For an arbitrary positive integer $n$, we use $[n]$ to represent the set $\{1, 2, \cdots, n\}$. We define $\mathbf{1}_n$ as a length-$n$ vector where all entries are ones. For any $x \in \mathbb{R}^n$, we use $\exp(x) \in \mathbb{R}^n$ to represent a length-$n$ vector whose $i$-th entry is $\exp(x_i)$. For any vector $x \in \mathbb{R}^n$, we use $x^\top$ to denote its transpose. For a vector $x$, the vector $\ell_2$ norm is denoted by $\|x\|_2$, i.e., $\|x\|_2 := (\sum_{i=1}^n x_i^2)^{1/2}$. For a vector $x$, we use $\|x\|_\infty$ to denote its $\ell_\infty$ norm, i.e., $\|x\|_\infty := \max_{i=1}^n |x_i|$. For a vector $x$, we use $\|x\|_1$ to denote its entrywise $\ell_1$ norm, i.e., $\|x\|_1 := \sum_{i=1}^n |x_i|$. For a matrix, we use $\|A\|_1$ to denote its $\ell_1$ norm, i.e., $\|A\|_1 = \sum_{j,l} |A_{j,l}|$. We use $\|A\|_\infty$ to denote its $\ell_\infty$ norm, i.e., $\|A\|_\infty := \max_{j,l} |A_{j,l}|$. For a vector $x \in \mathbb{R}^n$, we use $\mathrm{diag}(x)$ to denote a diagonal matrix where $i, i$-th entry on diagonal is $x_i$ for all $i \in [n]$.

**Definition 3.1.** *For a vector $x \in \mathbb{R}^n$, we define $\alpha(x) := \langle \exp(x), \mathbf{1}_n \rangle$. We define $\mathsf{softmax}(x)$ as $\mathsf{softmax}(x) := \alpha(x)^{-1} \exp(x)$. For a matrix $A$, we use the notation $\mathsf{softmax}(A)$ to denote that we apply $\mathsf{softmax}$ to each row of $A$ individually. We define $\mathsf{softm}(X, W_Q, W_K)$ as follows*

$$\mathsf{softm}(X, W_Q, W_K) := \mathsf{softmax}(X W_Q W_K^\top X^\top) = \begin{bmatrix} \mathsf{softmax}((X W_Q W_K^\top X^\top)_{1,*}) \\ \mathsf{softmax}((X W_Q W_K^\top X^\top)_{2,*}) \\ \vdots \\ \mathsf{softmax}((X W_Q W_K^\top X^\top)_{n,*}) \end{bmatrix}$$

*In many places, we will just write $\mathsf{softm}(X)$ for simplicity, and the weight matrices $W_Q, W_K$ will be clear from context.*

**Fact 3.2** (Shift-invariance property of softmax). *For any vector $x \in \mathbb{R}^n$ and for any fixed scalar $a \in \mathbb{R}$, we have $\mathsf{softmax}(x) = \mathsf{softmax}(x + a\mathbf{1}_n)$.*

**Fact 3.3** (Norm inequality). *For any matrices $A \in \mathbb{R}^{n \times d}, B \in \mathbb{R}^{d \times m}$ we have (1) $\|AB\|_1 \leq \|A\|_1 \cdot \|B\|_1$, (2)$\|AB\|_\infty \leq d\|A\|_\infty \cdot \|B\|_\infty$, (3) $\|AB\|_1 \leq m\|A\|_1 \cdot \|B\|_\infty$.*

### 3.2 DEFINITIONS OF Res

**Definition 3.4** (Res). *Let $Z \in \mathbb{R}^{n \times d}$ denote any matrix, we define function the Res $: \mathbb{R}^{n \times d} \to \mathbb{R}^{n \times d}$ as $\mathsf{Res}(Z) := Z - \mathbf{1}_n y^\top$ where $y := \arg\min_{y \in \mathbb{R}^d} \|Z - \mathbf{1}_n y^\top\|_\infty$.*

Res is the key definition behind the notion of rank collapse from prior work (Dong et al., 2021); we will use it here to study layer collapse as well, although we use the $\infty$ norm here in contrast to prior work which uses a $1, \infty$ norm.

### 3.3 $\theta$-BALANCE

We also need a measure of how balanced a matrix is.

**Definition 3.5** ($\theta$-balance). *Given a matrix $E \in \mathbb{R}^{n \times n}$, we define a corresponding matrix $D \in \mathbb{R}^{n \times n}$ to be the diagonal matrix with $D_{i,i} := \max_{j,l \in [n]} |E_{i,j} - E_{i,l}|$. We say $E$ is $\theta$-balanced, if $\|D\|_\infty \leq \theta$.*

### 3.4 SELF-ATTENTION NETWORK

**Definition 3.6.** *Let $g$ denote the entry-wise exponentiation function, i.e., for $z \in \mathbb{R}$ we have $g(z) = \exp(z)$, and for a matrix $W$ we have $g(W)_{i,j} = g(W_{i,j})$. Given $A \in \mathbb{R}^{n \times d}$ and weights $Q, K, V \in \mathbb{R}^{d \times d}$, the attention computation can be defined as*

$$\mathsf{SAtt}_H(X) := \sum_{h=1}^{H} \underbrace{D_h^{-1}}_{n \times n} \underbrace{g(X Q_h K_h^\top X^\top)}_{n \times n} \underbrace{X}_{n \times d} \underbrace{V_h}_{d \times d}$$

*where $D_h := \mathrm{diag}(g(X Q_h K_h^\top X^\top)\mathbf{1}_n)$, and where $\mathbf{1}_n \in \mathbb{R}^n$ is a length-$n$ vector whose entries are all $1$. When $H = 1$ we simply write $\mathsf{SAtt}$ to denote the $\mathsf{SAtt}_H$ function.*

**Definition 3.7.** *Let $L, H$ denote fixed constants, where $L$ represents the number of layers of the network, and $H$ represents the number of heads per layer. Let $\mathsf{SAtt}_H$ denote the multi-heads version of $\mathsf{SAtt}$ where $H$ is the number of heads. For each $\ell \in [L]$, $X_\ell \in \mathbb{R}^{n \times d}$ denote the $\ell$-th layer input of self-attention network, then we have $X_{\ell+1} = \mathsf{SAtt}_H(X_\ell) + X_\ell$.*

## 4 PERTURBATION PROPERTY

We now move on to our main proof of layer collapse. We begin by showing that the relevant measure of matrices to not change much when their inputs are perturbed. We will ultimately show that layer collapse occurs because lower layers of the network can be seen as slightly perturbing their inputs. We study the Res function in Section 4.1, the $\alpha$ function in Section 4.2.

### 4.1 PERTURBATION PROPERTY OF RES FUNCTION

**Lemma 4.1.** *Let $\mathsf{Res}()$ be defined as Def. 3.4. If $\|A - B\|_\infty \leq \epsilon$, then $\|\mathsf{Res}(A) - \mathsf{Res}(B)\|_\infty \leq 2\epsilon$.*

See Lemma E.1 in the Appendix for the proof of Lemma 4.1.

### 4.2 PERTURBATION PROPERTY OF EXP FUNCTION

**Lemma 4.2.** *Let $a, b \in \mathbb{R}^n$ such that $\|b\|_\infty \leq \epsilon$. Then, we can show: 1) $|\exp(a_i + b_i) - \exp(a_i)| \leq (e^\epsilon - 1) \cdot \exp(a_i)$. 2) $|\exp(a_i + b_i) - \exp(a_i)| \leq (e^\epsilon - 1) \cdot \exp(a_i + b_i)$. 3) $|\alpha(a+b) - \alpha(a)| \leq (e^\epsilon - 1) \cdot \alpha(a)$. 4) $|\alpha(a+b) - \alpha(a)| \leq (e^\epsilon - 1) \cdot \alpha(a+b)$.*

*Proof.* It is easy to see that

$$\max\{|\exp(-b_i) - 1|, |\exp(b_i) - 1|\} \leq e^\epsilon - 1 \tag{2}$$

We can show

$$|\exp(a_i + b_i) - \exp(a_i)| = \exp(a_i)|\exp(b_i) - 1| \leq \exp(a_i) \cdot (e^\epsilon - 1) \tag{3}$$

where the first step follows from simple algebra, the second step follows from Eq. (2).

Thus, we have

$$|\alpha(a+b) - \alpha(a)| = |\langle \exp(a+b), \mathbf{1}_n \rangle - \langle \exp(a), \mathbf{1}_n \rangle| \leq \sum_{i=1}^{n} |\exp(a_i + b_i) - \exp(a_i)|$$

$$\leq \sum_{i=1}^{n} \exp(a_i) \cdot (e^\epsilon - 1) = (e^\epsilon - 1)\alpha(a)$$

where the second step follows from triangle inequality, the third step follows from Eq. (3), the last step follows from definition of $\alpha(\cdot)$ function.

Similarly, we can show

$$|\exp(a_i + b_i) - \exp(a_i)| = \exp(a_i + b_i)|\exp(-b_i) - 1| \leq \exp(a_i + b_i) \cdot (e^\epsilon - 1) \quad (4)$$

where the first step follows from simple algebra, the second step follows from Eq. (2).

Then, we have

$$|\alpha(a+b) - \alpha(a)| = |\langle \exp(a+b), \mathbf{1}_n \rangle - \langle \exp(a), \mathbf{1}_n \rangle| \leq \sum_{i=1}^{n} |\exp(a_i + b_i) - \exp(a_i)|$$

$$\leq \sum_{i=1}^{n} \exp(a_i + b_i) \cdot (e^\epsilon - 1) = (e^\epsilon - 1)\alpha(a+b)$$

where the first step follows from definition of $\alpha$ (Definition 3.1), the second step follows from triangle inequality, the third step follows from Eq. (4), and the last step follows from definition of $\alpha$ (Definition 3.1). Thus, we complete the proof. $\square$

## 5 RANK COLLAPSE PROPERTY

In Section 5.1, we present a Lemma which connects Res(SAtt()) and Res(). In Section 5.2, we present our key lemma, a perturbation theorem for a layer of a Transformer. In Section 5.3, we present our main result and proof sketch.

### 5.1 THE CONNECTION BETWEEN Res(SAtt()) AND Res()

We next establish the relationship between Res(SAtt()) and Res() in terms of the balance of the inputs.

**Lemma 5.1.** *If the following conditions hold: Let $X \in \mathbb{R}^{n \times d}$ denote the input of attention layer. Let $\widetilde{X} = \mathsf{SAtt}(X)$ (see Definition 1.1 for function SAtt). Let $W_q, W_k, W_v \in \mathbb{R}^{d \times d}$ be the weight matrices of SAtt. Let $W = W_q W_k^\top$. Let $E = \beta \, \mathsf{Res}(X) W \, \mathsf{Res}(X)^\top$. Suppose that $E$ is a $\theta$-balanced matrix (see Definition 3.5). Let $\beta := 1/\sqrt{d_0}$ denote the normalization factor. Let $K := (e^\theta - 1)\|W_v\|_\infty$. Then, we have $\|\mathsf{Res}(\mathsf{SAtt}(X))\|_\infty \leq K \cdot \|\mathsf{Res}(X)\|_\infty$.*

*Proof.* The unscaled attention scores are computed as follows $A = (XW_q + \mathbf{1}_n b_q^\top) \cdot (XW_k + \mathbf{1}_n b_k^\top)^\top$. Recall that $W = W_q W_k^\top$. For notational convenience, we define $b := W_k b_q$.

We can use the softmax shift invariance property to remove terms which are constant over the columns and obtain, $A = \underbrace{X}_{n \times d} \underbrace{W}_{d \times d} \underbrace{X^\top}_{d \times n} + \underbrace{\mathbf{1}_n}_{n \times 1} \underbrace{b^\top}_{1 \times d} \underbrace{X^\top}_{d \times n}$.

We define $\widetilde{R} := \mathsf{Res}(\widetilde{X}) \in \mathbb{R}^{n \times d}$ (Recall the definition of the function Res() in Definition 3.4).

In next equation, we will use the definition of $R$ to simplify $A$. The attention matrix can be written as

$$A = \beta \cdot (\mathbf{1}_n x^\top + R) W (\mathbf{1}_n x^\top + R)^\top + \beta \cdot \mathbf{1}_n b^\top (\mathbf{1}_n x^\top + R)^\top$$

$$= \beta \cdot (x^\top W x \mathbf{1}_n + RWx + \mathbf{1}_n b^\top x)\mathbf{1}_n^\top + \beta \cdot (RWR^\top + \mathbf{1}_n x^\top WR^\top + \mathbf{1}_n b^\top R^\top) \quad (5)$$

Using Fact 3.2, we can remove the first term in the above equation since it is constant across columns. We thus have that the following equation for $P = \mathsf{softmax}(A) \in \mathbb{R}^{n \times n}$

$$P = \mathsf{softmax}(\beta RWR^\top + \mathbf{1}_n r^\top) = \mathsf{softmax}(E + \mathbf{1}_n r^\top) \quad (6)$$

where the first step follows from $r = \beta R(W^\top x + b) \in \mathbb{R}^n$, the second step follows from setting $E = \beta RWR^\top \in \mathbb{R}^{n \times n}$.

To continue the proof, we also set $\widetilde{A} = \mathbf{1}_n r^\top \in \mathbb{R}^{n \times n}$, the input reweighted by the attention probabilities $PX$ will be entry-wisely upper bounded as follows

$$
\begin{aligned}
PX = P(\mathbf{1}_n x^\top + R) &= \mathbf{1}_n x^\top + PR \\
&= \mathbf{1}_n x^\top + \mathsf{softmax}(\mathbf{1}_n r^\top + E)R \\
&\leq \mathbf{1}_n x^\top + (I + e^D - I)\mathbf{1}_n \mathsf{softmax}(r)^\top R \\
&= \mathbf{1}_n(x^\top \mathsf{softmax}(r)^\top R) + (e^D - I)\mathbf{1}_n \mathsf{softmax}(r)^\top R \quad (7)
\end{aligned}
$$

where the first step follows from definition of $R$, the second step follows from $P\mathbf{1}_n = \mathbf{1}_n$, the third step follows from Eq. (6) and $e^D$ is a diagonal matrix, the forth step follows from Lemma B.3 and $e^D$ is diagonal matrix where the $i, i$-th entry on diagonal is $e^{D_{i,i}}$ (see Definition 3.5 for $D$, $D_{i,i} \geq 0$).

Therefore, the entry-wise distance of the output of the self-attention layer $\mathsf{SAtt}(X) = PXW_v$ from being constant across token is at most:

$$|[\mathsf{SAtt}(X) - \mathbf{1}_n \widetilde{r}^\top]_{i,j}| = |[PXW_v - \mathbf{1}_n \widetilde{r}^\top]_{i,j}| \leq (e^\theta - 1) \cdot |[\mathbf{1}_n \mathsf{softmax}(r)^\top RW_v]_{i,j}|$$

where the second step follows from $\widetilde{r} = (x + R^\top \mathsf{softmax}(r))W_v$, Eq. (7), and $\theta$ (see Definition 3.5).

Now we bound the right hand side of the above inequality.

For $\| \cdot \|_\infty$, we can show

$$
\begin{aligned}
\| \mathsf{SAtt}(X) - \mathbf{1}_n \widetilde{r}^\top \|_\infty &\leq (e^\theta - 1)\|\mathbf{1}_n\|_\infty \cdot \|\mathsf{softmax}(r)^\top RW_v\|_\infty \\
&\leq (e^\theta - 1)\|\mathbf{1}_n\|_\infty \|R\|_\infty \|W_v\|_\infty \leq (e^\theta - 1)\|R\|_\infty \cdot \|W_v\|_\infty, \quad (8)
\end{aligned}
$$

where the last step follows from Definition 3.5.

Note that $R' = \mathsf{Res}(\mathsf{SAtt}(X))$ and $R = \mathsf{Res}(X)$ and using the definition of $K$ in Lemma statement, we can show $\| \mathsf{Res}(\mathsf{SAtt}(X))\|_\infty \leq K \cdot \| \mathsf{Res}(X)\|_\infty$. Thus, we complete the proof. $\qquad \square$

## 5.2 Perturbation of One Transformer Layer

We next give a toy lemma to demonstrate the idea behind our approach. We will assume for simplicity here that $d = n$ so $X$ is a square matrix, and we focus on a layer with only one attention head. The full proof for any $d \neq n$ and multiple heads appears in Lemma C.1 in the Appendix below.

**Lemma 5.2** (Single Head). *Let $n = d$ and let $X \in \mathbb{R}^{n \times d}$. Let $A = \mathsf{softm}_1(X)$ (Recall that $\mathsf{softm}()$ function is defined as Definition 3.1. Note that $\mathsf{softm}_1$ and $\mathsf{softm}_2$ are two different instantiations with different $W_k, W_q, W_v$ weights with $\|W_k\|_\infty, \|W_q\|_\infty, \|W_v\|_\infty \leq \eta$. Let $B = X + A$. Suppose $\| \mathsf{Res}(A)\|_\infty \leq K \cdot \| \mathsf{Res}(X)\|_\infty \leq \epsilon$. (We remark that this condition will hold due to Lemma 5.1; here $K$ is as defined in Lemma 5.1). Let $g(\epsilon) := 4\epsilon$ and let $\epsilon_0 = 3g(3\epsilon)$. Then we can show*

$$\| \mathsf{softm}_2(B) - \mathsf{softm}_2(X)\|_\infty \leq \epsilon_0.$$

*Proof.* Let $R_X = \mathsf{Res}(X)$ so that $X = R_X + \mathbf{1}_n y_X^\top$ for some vector $y_X \in \mathbb{R}^d$. Using Lemma D.6

$$\| \mathsf{softm}_2(X) - \mathsf{softm}_2(R_X)\|_\infty \leq g(\epsilon) \quad (9)$$

Let $R_A = \mathsf{Res}(A)$ so that $A = R_A + \mathbf{1}_n y_A^\top$ for some vector $y_A \in \mathbb{R}^d$. Let $R_B = \mathsf{Res}(B)$ so that $B = R_B + \mathbf{1}_n y_B^\top$ for some vector $y_B \in \mathbb{R}^d$. Using Lemma D.6, we can show that

$$\| \mathsf{softm}_2(B) - \mathsf{softm}_2(R_B)\|_\infty \leq g(\epsilon). \quad (10)$$

Thus, as long as $\|R_A\|_\infty \le \epsilon$, then using Lemma D.6, we have

$$\| \operatorname{softm}_2(R_X + R_A) - \operatorname{softm}_2(R_X)\|_\infty \le g(\epsilon) \tag{11}$$

We can show $R_X = \operatorname{Res}(X) = \operatorname{Res}(B - A) = \operatorname{Res}(B - R_A)$. Then, we know $\|R_X - R_B\|_\infty \le \| \operatorname{Res}(B - R_A) - \operatorname{Res}(B)\|_\infty \le 2\|R_A\|_\infty \le 2\epsilon$. Here, the second step follows from Lemma 4.1.

Recall $B = X + A$, and $\|R_A\|_\infty \le \epsilon$ then we know

$$\|R_X + R_A - R_B\|_\infty \le \|R_X - R_B\|_\infty + \|R_A\|_\infty \le 3\|R_A\|_\infty \le 3\epsilon.$$

Since $\|R_X + R_A - R_B\|_\infty \le 3\epsilon$, then using Lemma D.6, we have

$$\| \operatorname{softm}_2(R_X + R_A) - \operatorname{softm}_2(R_B)\|_\infty \le g(3\epsilon) \tag{12}$$

Then, we can show

$$\begin{aligned}
&\| \operatorname{softm}_2(B) - \operatorname{softm}_2(X)\|_\infty \\
=\ & \| \operatorname{softm}_2(B) - \operatorname{softm}_2(R_X)\|_\infty + g(\epsilon) \\
=\ & \| \operatorname{softm}_2(R_B) - \operatorname{softm}_2(R_X)\|_\infty + +2g(\epsilon) \\
\le\ & \| \operatorname{softm}_2(R_B) - \operatorname{softm}_2(R_X + R_A)\|_\infty + \| \operatorname{softm}_2(R_X + R_A) - \operatorname{softm}_2(R_X)\|_\infty + 2g(\epsilon) \\
\le\ & g(3\epsilon) + 3g(\epsilon) \le 3g(3\epsilon)
\end{aligned}$$

where the first step follows from triangle inequality and Eq. (9), the second step follows from triangle inequality and Eq. (10), the third follows from triangle inequality, the forth step follows from Eq. (11) and Eq. (12), and the last step follows from $g$ is monotone. $\square$

### 5.3 Putting it All Together

*Proof Sketch of Theorem 1.2.* We'll show what to do to delete one layer, then repeat that $L - 1$ times to get down to one layer. When we delete the first layer, Lemma C.1 (which is the version of Lemma 5.2 which deals with multiple heads) says that the output of the second layer will differ by at most $O(\eta \cdot \epsilon_0)$, where $\epsilon_0 = O(1) \cdot \|X\|_\infty$ is the constant from Lemma 5.1 and Lemma 5.2, and $X$ is the input of first layer of network. Therefore, by applying Lemma B.2 iteratively to each layer, it follows that the outputs of all subsequent layers will also change by at most $O(\eta \cdot \epsilon_0)$. In particular, the final output will differ by at most $O(\eta \cdot \epsilon_0)$. We finally repeat this $L - 1$ times to remove all but one layer and get the final error. We defer further proof details to the Appendix due to space limitations. $\square$

## 6 Conclusion

We have shown that Self Attention Networks must experience layer collapse unless they have large attention weights, even if they have skip connections. Our result proves that two different common notions in the literature are actually misconceptions.

The first misconception is the common interpretation of the prior work (Dong et al., 2021) that skip connections are the key to the expressive power of Self Attention Networks. We extend their result and show that even with skip connections, large weights are needed to prevent layer collapse.

The second misconception is that Self Attention Networks with smaller weights may still have reasonable expressive power. Indeed, although it is intuitive that bounding the magnitudes of weights must limit the expressive power to some extent, there is nonetheless a long line of work on trying to use networks with small weights, weight quantization, or similar approaches. This work is (presumably) hoping that the limit is only modest. We show that the limit is severe: networks with small weights cannot take advantage of more than one layer! This is the first theoretical limitation result on networks with small weights to our knowledge.

ETHIC STATEMENT

This paper does not involve human subjects, personally identifiable data, or sensitive applications. We do not foresee direct ethical risks. We follow the ICLR Code of Ethics and affirm that all aspects of this research comply with the principles of fairness, transparency, and integrity.

REPRODUCIBILITY STATEMENT

We ensure reproducibility of our theoretical results by including all formal assumptions, definitions, and complete proofs in the appendix. The main text states each theorem clearly and refers to the detailed proofs. No external data or software is required.

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

# Appendix

**Roadmap.** In Section A, we provide several simple definitions. In Section B, we show more perturbation properties for the softmax matrix. In Section C, we provide the proofs related to network layers with multiple attention heads. In Section D, we prove our main Theorem. In Section F, we provide the broader impact of this work. In Section G, we discuss the LLM usage.

## A   PRELIMINARIES

Note that $\mathsf{softm}(X)$ function defined above is ignoring the effect of the weights $W_v$. Here we incorporate them in another function $\mathsf{softmv}(X)$ (which is also usually called self-attention).

**Definition A.1.** *Let $W_q, W_k$ be weights being used in* $\mathsf{softm}$. *Let $W_v$ denote the extra weights that will be used in* $\mathsf{softmv}$. *We define* $\mathsf{softmv}(X)$ *as follows*

$$\mathsf{softmv}(X) := \mathsf{softm}(X)XW_v.$$

*In particular,*

$$\mathsf{softm}(X; W_Q, W_K) := D^{-1}\exp(XW_QW_K^\top X^\top)$$

*where $D$ is diagonal matrix $D = \mathrm{diag}(\exp(XW_QW_K^\top X^\top)\mathbf{1}_n)$.*

*For notational convenience, we will omit $W_Q, W_K$ when they are clear from context and simply write* $\mathsf{softm}(X)$.

Next we define a very useful parameter $\epsilon_\ell$ which captures the Lipschitz and layer norm property of every layer.

**Definition A.2.** *Let all layers' weights are bounded, i.e, $\|W_q\|_\infty, \|W_k\|_\infty, \|W_v\|_\infty \le \eta$. Let $X_0$ denote the first layer input of entire neural network and it is bounded $\|X_0\|_\infty \le \phi_0$. Let $H$ denote the number of heads. For each layer $\ell \in [L]$, we define a parameter $\epsilon_\ell := 2\eta\phi_0(1 + dH\eta)^\ell nd$. We will select parameters to enforce that $\phi_0 \le \epsilon_\ell$ in order to simplify some calculations below.*

**Definition A.3.** *We define function $g(\epsilon) := 4\epsilon$.*

## B   PERTURBATION PROPERTY OF SOFTMAX MATRIX

**Lemma B.1.** *If the following conditions hold: Let $a, b \in \mathbb{R}^n$. Let $\|b\|_\infty \le \epsilon$. Then we can show*

- $|\alpha(a+b)^{-1} - \alpha(a)^{-1}| \le (e^\epsilon - 1)\alpha(a)^{-1}$

- $|\alpha(a+b)^{-1} - \alpha(a)^{-1}| \le (e^\epsilon - 1)\alpha(a+b)^{-1}$

*Proof.* We can show that

$$
\begin{aligned}
|\alpha(a+b)^{-1} - \alpha(a)^{-1}| &= \alpha(a+b)^{-1}\alpha(a)^{-1}|\alpha(a+b) - \alpha(a)| \\
&\le \alpha(a+b)^{-1}\alpha(a)^{-1} \cdot (e^\epsilon - 1)\alpha(a+b) \\
&= (e^\epsilon - 1)\alpha(a)^{-1}
\end{aligned}
$$

where the first step follows from simple algebra, the second step follows from Lemma 4.2.

Similarly, we can also show $|\alpha(a+b)^{-1} - \alpha(a)^{-1}| \le (e^\epsilon - 1)\alpha(a+b)^{-1}$.  □

**Lemma B.2.** *Let* $\mathsf{softmax}$ *function be defined as Definition 3.1. Let $a, b \in \mathbb{R}^n$. If $|b_i| \le \epsilon$ for all $i \in [n]$, then, we can show that*

$$\|\mathsf{softmax}(a+b) - \mathsf{softmax}(a)\|_\infty \le 2(e^\epsilon - 1)$$

*Proof.* For each $i \in [n]$, we can show

$$|\alpha(a+b)^{-1}\exp((a+b)_i) - \alpha(a)^{-1}\exp(a_i)|$$

$$= |\alpha(a + b)^{-1} \exp((a + b)_i) - \alpha(a + b)^{-1} \exp(a_i) + \alpha(a + b)^{-1} \exp(a_i) - \alpha(a)^{-1} \exp(a_i)|$$

$$\leq \alpha(a + b)^{-1} |\exp(b_i + a_i) - \exp(a_i)| + \exp(a_i) \cdot |\alpha(a + b)^{-1} - \alpha(a)^{-1}|$$

$$:= A_1 + A_2$$

where the second step follows from the triangle inequality.

We can upper bound $A_1$ as

$$A_1 = \alpha(a + b)^{-1} \cdot |\exp(a_i + b_i) - \exp(a_i)|$$

$$\leq \alpha(a + b)^{-1} \cdot (e^\epsilon - 1) \exp(a_i + b_i)$$

$$\leq (e^\epsilon - 1)$$

where the second step follows from Lemma 4.2, the third step follows from $\alpha(x)^{-1} \exp(x_i) \in (0, 1)$ for any $x$ and $i$.

We can upper bound $A_2$ as

$$A_2 = \exp(a_i) \cdot |\alpha(a + b)^{-1} - \alpha(a)^{-1}|$$

$$\leq \exp(a_i) \cdot (e^\epsilon - 1)\alpha(a)^{-1}$$

$$\leq e^\epsilon - 1$$

where the second step follows from Lemma B.1, and the third step follows from $\alpha(x)^{-1} \exp(x_i) \in (0, 1)$ for any $x$ and $i$.

Putting everything together, we can show

$$\|\mathsf{softmax}(a + b) - \mathsf{softmax}(a)\|_\infty = \max_{i \in [n]} |\alpha(a + b)^{-1}(a + b)_i - \alpha(a)^{-1} a_i|$$

$$\leq 2(e^\epsilon - 1).$$

Thus, we complete the proof. $\qquad\square$

**Lemma B.3.** *If the following conditions hold*

- *Let $P = \mathsf{softmax}(A)$ (see Definition 3.1 for function $\mathsf{softmax}()$).*

- *Let $\widetilde{A} = A - E$.*

- *Let $\widetilde{P} = \mathsf{softmax}(\widetilde{A})$.*

- *Let $D$ be defined as Definition 3.5, i.e., $D_{i,i} := \max_{j,l \in [n]} |E_{i,j} - E_{i,l}|$*

*Then we can show, for all $i \in [n], j \in [n]$*

$$e^{-D_{i,i}} \widetilde{P}_{i,j} \leq P_{i,j} \leq e^{D_{i,i}} \widetilde{P}_{i,j}.$$

*Proof.* Let us start by the definition of $P$, for each $i \in [n], j \in [n]$

$$P_{i,j} = (\mathsf{softmax}(A))_{i,j}$$

$$= (\mathsf{softmax}(\widetilde{A} + E))_{i,j}$$

$$= \frac{\exp(\widetilde{A}_{i,j} + E_{i,j})}{\sum_{l=1}^n \exp(\widetilde{A}_{i,l} + E_{i,l})}$$

$$= \frac{\exp(\widetilde{A}_{i,j}))}{\sum_{l=1}^n \exp(\widetilde{A}_{i,l}) \exp(E_{i,l} - E_{i,j})}$$

where the first step follows from definition of $P$, the second step follows from definition of $A$, the third step follows from definition of softmax (Definition 3.1), and the last step follows from property of exp.

We define $D_{i,i} := \max_{j,l \in [n]} |E_{i,j} - E_{i,l}|$. We have that

$$P_{i,j} \in [\widetilde{P}_{i,j} \exp(-D_{i,i}), \widetilde{P}_{i,j} \exp(D_{i,i})]$$

Thus, we complete the proof. $\qquad\square$

**Lemma B.4.** *If the following conditions hold*

- *Let $W_q, W_k, W_v$ be the matrix that $\|W_q\|_\infty, \|W_k\|_\infty, \|W_v\|_\infty \leq \eta$.*

- *Let $W = W_q W_k^\top$.*

- *Let $E = \beta \, \mathsf{Res}(X) W \, \mathsf{Res}(X)^\top$.*

- *Let $\beta$ satisfy that $\beta \leq 1/(\|\mathsf{Res}(X)\|_\infty^2 \eta^2)$.*

*Then, we have $E$ is $\theta$-balanced with $\theta = 1$.*

*Proof.* First, note that $\|W\|_\infty \leq \|W_q\|_\infty \cdot \|W_k\|_\infty \leq \eta^2$.
We can show

$$
\begin{aligned}
\|E\|_\infty &\leq \beta \cdot \|\mathsf{Res}(X)\|_\infty^2 \cdot \|W\|_\infty \\
&\leq \beta \cdot \|\mathsf{Res}(X)\|_\infty^2 \cdot \eta^2 \\
&\leq 1
\end{aligned}
$$

Thus, we complete the proof. $\qquad\square$

## C  MULTIPLE HEADS

Here, we generalize the proof of Lemma 5.2 to multiple heads. Note that Lemma 5.2 presented a simplified proof by ignoring the effects of $XW_v$, and thus automatically assuming $n = d$. In this section we remove that condition and prove the result for general $n$ and $d$. In Section C.1, our goal is to prove Lemma C.1, which is the multiple heads version of Lemma 5.2. In Section C.2, we show that several required conditions in Lemma C.1 are satisfied.

### C.1  MULTIPLE HEADS FOR SKIPPING ONE LAYER

In the next Lemma C.1, we will put the effect of softmv back. We remark that the major idea of the proof remains the same as Lemma 5.2.

**Lemma C.1** (Multiple Heads version of Lemma 5.2)**.** *If the following conditions hold,*

- *Let $H$ denote the number of heads.*

- *Note that $\mathsf{softm}_1$ and $\mathsf{softm}_2$ are two different instantiations with different $W_k, W_q, W_v$ weights.*

- *Let $X \in \mathbb{R}^{n \times d}$.*

- *$A_i = \mathsf{softmv}_{1,i}(X) \in \mathbb{R}^{n \times d}$ for $i \in [H]$. (Let $\mathsf{softmv}$ be defined as Definition A.1)*

- *$B = X + \sum_{i=1}^H A_i \in \mathbb{R}^{n \times d}$.*

- *$\|\mathsf{Res}(A_i)\|_\infty \leq K \cdot \|\mathsf{Res}(X)\|_\infty \leq \epsilon$ for all $i \in [H]$. (We remark that this condition will hold due to Lemma 5.1; here $K$ is as defined in Lemma 5.1)*

- *Let $g(\epsilon) := 4\epsilon$ (see Definition A.3).*

- *Let $W_v$ satisfy that $\|(B - X)W_v\|_\infty \leq \epsilon$ and $n\|XW_v\|_\infty \leq \epsilon \leq 1$ (These conditions will be verified by Lemma C.2).*

- *Let $\epsilon_0 = 3g(3H\epsilon)$.*

*Then we can show*

- **Part 1.** $\|\mathsf{softm}_2(B) - \mathsf{softm}_2(X)\|_\infty \leq \epsilon_0$

- **Part 2.** $\|\mathsf{softmv}_2(B) - \mathsf{softmv}_2(X)\|_\infty \leq \epsilon_0$

*Proof.* **Proof of Part 1.**

Let $R_X = \text{Res}(X)$ so that $X = R_X + \mathbf{1}_n y_X^\top$ for some vector $y_X \in \mathbb{R}^d$.

Using Lemma D.6, we can show that

$$\| \text{softm}_2(X) - \text{softm}_2(R_X) \|_\infty \leq g(\epsilon). \tag{13}$$

To notataionaly help in our proof, we define the prefix sums of matrices $A_0, A_1, \cdots, A_i \in \mathbb{R}^{n \times d}$ as

$$A_{[i]} := \sum_{j=0}^{i} A_i$$

where $A_0$ is an artificial matrix that has 0 everywhere.

For each $i \in [H]$, let $R_{A_{[i]}} = \text{Res}(A_{[i]})$ so that $A_{[i]} = R_{A_{[i]}} + \mathbf{1}_n y_{A_{[i]}}^\top$ for some vector $y_{A,[i]} \in \mathbb{R}^d$.

Using Lemma D.6, we can show that

$$\| \text{softm}_2(A_{[i]}) - \text{softm}_2(R_{A_{[i]}}) \|_\infty \leq g(\epsilon).$$

Let $R_B = \text{Res}(B)$ so that $B = R_B + \mathbf{1}_n y_B^\top$ for some vector $y_B \in \mathbb{R}^d$. Using Lemma D.6, we can show that

$$\| \text{softm}_2(B) - \text{softm}_2(R_B) \|_\infty \leq g(\epsilon) \tag{14}$$

Since $\|R_{A_{[i]}} - R_{A_{[i-1]}}\|_\infty \leq \epsilon$ for all $i \in [H]$, then using Lemma D.6 , we have: for each $i \in [H]$

$$\| \text{softm}_2(R_X + R_{A_{[i]}}) - \text{softm}_2(R_X + R_{A_{[i-1]}}) \|_\infty \leq g(\epsilon) \tag{15}$$

We can show $R_X = \text{Res}(X) = \text{Res}(B - A_{[H]}) = \text{Res}(B - R_{A_{[H]}})$.

Then, we know

$$\|R_X - R_B\|_\infty = \| \text{Res}(B - R_{A_{[H]}}) - \text{Res}(B) \|_\infty$$
$$\leq 2\|R_{A_{[H]}}\|_\infty \tag{16}$$

where the first step follows from $R_X = \text{Res}(B - R_{A_{[H]}})$ and $R_B = \text{Res}(B)$, the second step follows from Lemma 4.1.

Recall $B = X + A$, and $\|R_A\|_\infty \leq \epsilon$ then we know

$$\|R_X + R_A - R_B\|_\infty \leq \|R_X - R_B\|_\infty + \|R_{A_{[H]}}\|_\infty$$
$$\leq 3\|R_{A_{[H]}}\|_\infty$$
$$\leq 3H\epsilon,$$

where the first step follows from triangle inequality, the second step follows from $\|R_X - R_B\|_\infty \leq 2\|R_{A_{[H]}}\|_\infty$, and the last step follows from $\|R_{A_{[H]}}\|_\infty \leq H\epsilon$.

Since $\|R_X + R_{A,H} - R_B\|_\infty \leq 3H\epsilon$, then using Lemma D.6 , we have

$$\| \text{softm}_2(R_X + R_{A,H}) - \text{softm}_2(R_B) \|_\infty \leq g(3H\epsilon) \tag{17}$$

Then, we can show

$$\| \text{softm}_2(B) - \text{softm}_2(X) \|_\infty$$
$$\leq \| \text{softm}_2(B) - \text{softm}_2(R_X) \|_\infty + g(\epsilon)$$
$$\leq \| \text{softm}_2(R_B) - \text{softm}_2(R_X) \|_\infty + 2g(\epsilon)$$
$$\leq \| \text{softm}_2(R_B) - \text{softm}_2(R_X + R_{A,H}) \|_\infty$$
$$+ \sum_{i=1}^{H-1} \| \text{softm}_2(R_X + R_{A,i}) - \text{softm}_2(R_X + R_{A,i-1}) \|_\infty + 2g(\epsilon)$$

$$\leq g(3H\epsilon) + (H+2) \cdot g(\epsilon) \tag{18}$$
$$\leq 2g(3H\epsilon)$$

where the first step follows from triangle inequality and Eq. (13), the second step follows from triangle inequality and Eq. (14), the third step follows from triangle inequality, the forth step follows from Eq. (15) and Eq. (17), and the last step follows from property of function $g$.

**Proof of Part 2.**

We can show that

$$\| \operatorname{softmv}_2(B) - \operatorname{softmv}_2(X) \|_\infty$$
$$= \| \operatorname{softm}_2(B)BW_v - \operatorname{softm}_2(X)XW_v \|_\infty$$
$$\leq \| \operatorname{softm}_2(B)BW_v - \operatorname{softm}_2(B)XW_v \|_\infty + \| \operatorname{softm}_2(B)XW_v - \operatorname{softm}_2(X)XW_v \|_\infty$$
$$\leq \|(B-X)W_v\|_\infty + \| \operatorname{softm}_2(B) - \operatorname{softm}_2(X) \|_\infty \cdot n\|XW_v\|_\infty$$
$$\leq \epsilon + 2g(3H\epsilon) \cdot n\|XW_v\|_\infty$$
$$\leq 3g(3H\epsilon)$$

where the second step follows from triangle inequality, the third step follows from Fact 3.3 and Fact E.2, and the forth step follow from Eq. (18) , where the last step follows from property of function $g$ and assumption in Lemma statement. $\square$

### C.2 Conditions in Lemma C.1 are Satisfied

Here we will show that the three conditions in Lemma C.1 will be satisfied for each layer $\ell$.

- $\| \operatorname{Res}(A_i)\|_\infty \leq K \cdot \| \operatorname{Res}(X)\|_\infty \leq \epsilon$ (where $K := (e^\theta - 1)\|W_v\|_\infty$, definition of $K$ recall Lemma 5.1). Here $\theta = 1$ due to Lemma B.4
- $\|(B-X)W_v\|_\infty \leq \epsilon$
- $n\|XW_v\|_\infty \leq \epsilon$

**Lemma C.2.** *If the following conditions hold*

- $\epsilon_\ell := 2\eta\phi_0(1 + H\eta)^\ell nd.$ *(see Definition A.2)*
- *Let $\eta \in (0, 1]$.*
- *Let $\epsilon_\ell \in (0, 1)$.*

*Then, we can show*

- *Part 1. $K \cdot \| \operatorname{Res}(X_\ell)\|_\infty \leq \epsilon_\ell$*
- *Part 2. $\|(B-X_\ell)W_v\|_\infty \leq \epsilon_\ell$*
- *Part 3. $n\|X_\ell W_v\|_\infty \leq \epsilon_\ell$*

*Proof.* **Proof of Part 1.**

We can show that

$$K \cdot \| \operatorname{Res}(X_\ell)\|_\infty \leq 2\eta\| \operatorname{Res}(X_\ell)\|_\infty$$
$$\leq 2\eta\|X_\ell\|_\infty$$
$$\leq 2\eta\phi_0 \cdot (1 + dH\eta)^\ell$$
$$= \epsilon_\ell$$

where the first step follows from $\theta = 1$ and $\|W_v\|_\infty \leq \eta$, the third step follows from Lemma D.8, and the last step follows from definition $\epsilon_\ell$

**Proof of Part 2.**

$$\|(B-X_\ell)W_v\|_\infty \leq \eta \cdot d\|B - X_\ell\|_\infty$$

$$= \eta \cdot d \sum_{i=1}^{H} \| \operatorname{softmv}_i(X_\ell) \|_\infty$$

$$\leq \eta \cdot d \sum_{i=1}^{H} \| \operatorname{softm}_i(X_\ell) X_\ell W_{v,i} \|_\infty$$

$$\leq \eta \cdot d \sum_{i=1}^{H} \| X_\ell W_{v,i} \|_\infty$$

$$\leq \eta \cdot dH \cdot \| X_\ell W_{v,i} \|_\infty$$

$$\leq \eta \cdot dH \cdot \epsilon_\ell$$

$$\leq \epsilon_\ell$$

where the first step follows from $\|W_v\|_\infty \leq \eta$, the second step follows from definition of $B$, the third step follows from definition of softmv, the forth step follows from Fact E.2, the sixth step follows from part 3, and the last step follows from $\eta \leq 1/(dH)$.

**Proof of Part 3.**

We can show

$$\| X_\ell W_v \|_\infty \leq \eta d \| X_\ell \|_\infty$$

$$\leq \eta d \phi_0 (1 + dH\eta)^\ell$$

$$\leq \epsilon_\ell.$$

where the second step follows from Lemma D.8, the third step follows from choice of $\epsilon_\ell$.

$\square$

# D MULTIPLE LAYERS

In Section D.1, we provide the proof of our main theorem. In Section D.2, we provide the Lipshitz property of several key functions being used in our proofs. In Section D.3, we prove the Lipschitz property for each layer of our Self Attention Network. Finally, in Section D.4, prove the norm of each layer in the Self Attention Network is not increasing much.

## D.1 PROOF OF THEOREM D.1

**Theorem D.1** (Formal version of Theorem 1.2). *Suppose $S$ is a* SAtt *with residuals, with the property that for every attention head in every one of its layers, the weight matrices $W_q, W_k, W_v \in \mathbb{R}^{d \times d}$ all have the bound $\|W_q\|_\infty, \|W_k\|_\infty, \|W_v\|_\infty \leq \eta$. Let $H$ denote the number of heads. Let $L$ denote the number of layers. Assume $\eta \leq A \cdot \min\{1/(HLd), 1/(\phi_0 nd)\} \leq 1$ for some parameter $A \leq O(1)$. Then, there exists a* SAtt *with residuals $S'$ with just one layer so that, for any bounded $X \in \mathbb{R}^{n \times d}$ with $\|X\|_\infty \leq \phi_0$, we have $\|S(X) - S'(X)\|_\infty \leq O(A/L)$.*

*Proof.* We define

$$X_\ell^{\ell_0} = B_\ell^{\ell_0}$$

Then, we define

$$B_\ell^{\ell_0} = \begin{cases} X_{\ell-1}^{\ell_0} + \sum_{i=1}^{H} A_{\ell-1,i}^{\ell_0}, & \text{if } \ell \leq \ell_0; \\ \sum_{i=1}^{H} A_{\ell-1,i}^{\ell_0} & \text{otherwise.} \end{cases}$$

Let softmv() function be defined as Defintion A.1. We define

$$A_{\ell-1,i}^{\ell_0} = \operatorname{softmv}_{\ell-1,i}^{\ell_0}(X_{\ell-1}^{\ell_0})$$

Note that the notation $B_L^0$ means we have residual in every layer, whereas the notation $B_L^{\ell_0}$ means we don't have a residual connection from layer $\ell_0$ to layer $L$.

Let $\epsilon_\ell$ be defined as Definition A.2. Let $\delta := \max_{\ell \in [L]} 3g(3H\epsilon_\ell)$. Using Lemma C.1, we can show for all $\ell \in [L]$,

$$\| \mathsf{softmv}_\ell(B_\ell^{\ell-1}) - \mathsf{softmv}_\ell(B_\ell^\ell) \|_\infty \le \delta$$

Let $C := \max_{\ell \in [L]} 3d\eta(\epsilon_\ell^2 + 1)$.

Then we can show

$$
\begin{aligned}
&\| \mathsf{softmv}_2(B_2^0) - \mathsf{softmv}_2(B_2^2) \|_\infty \\
&\le \| \mathsf{softmv}_2(B_2^0) - \mathsf{softmv}_2(B_2^1) \|_\infty + \| \mathsf{softmv}_2(B_2^1) - \mathsf{softmv}_2(B_2^2) \|_\infty \\
&\le C \cdot \| \mathsf{softmv}_1(B_1^0) - \mathsf{softmv}_1(B_1^1) \|_\infty + \| \mathsf{softmv}_2(B_2^1) - \mathsf{softmv}_2(B_2^2) \|_\infty \\
&\le (C+1)\delta
\end{aligned}
$$

where the first step follows from triangle inequality, the second step follows from the fact that one layer of the network is $C$-Lipschitz (see Lemma D.6), and the last step follows from merging the errors.

For three layers, we have

$$
\begin{aligned}
&\| \mathsf{softmv}_3(B_3^0) - \mathsf{softmv}_3(B_3^3) \|_\infty \\
&\le \| \mathsf{softmv}_3(B_3^0) - \mathsf{softmv}_3(B_3^1) \|_\infty \\
&\quad + \| \mathsf{softmv}_3(B_3^1) - \mathsf{softmv}_3(B_3^2) \|_\infty \\
&\quad + \| \mathsf{softmv}_3(B_3^2) - \mathsf{softmv}_3(B_3^3) \|_\infty \\
&\le C^2 \cdot \| \mathsf{softmv}_1(B_1^0) - \mathsf{softmv}_1(B_1^1) \|_\infty \\
&\quad + C \cdot \| \mathsf{softmv}_2(B_2^1) - \mathsf{softmv}_2(B_2^2) \|_\infty \\
&\quad + \| \mathsf{softmv}_3(B_3^2) - \mathsf{softmv}_3(B_3^3) \|_\infty \\
&\le C^2\delta + C\delta + \delta \\
&= (C^2 + C + 1)\delta
\end{aligned}
$$

where the first step follows from triangle inequality, the second step follows from one layer of network is $C$-Lipshitz (see Lemma D.6), and the forth step follows from Lemma C.1, and the last step follows from merging the errors.

Therefore for $L$ layers we have

$$\| \mathsf{softmv}_L(B_L^0) - \mathsf{softmv}_L(B_L^L) \|_\infty \le (C^L + \cdots + C + 1)\delta$$

Thus we complete the proof.

Now, we are ready to analyze the final bound, recall that above we have $\epsilon_\ell = 2\phi_0\eta nd(1 + dH\eta)^\ell \in (0,1)$, $\delta = \max_\ell 3g(3H\epsilon_\ell)$, and $C = \max_\ell 3d\eta(\epsilon_\ell^2 + 1)$.

Recall that $\eta \le A \cdot \min\{1/(HLd), 1/(\phi_0 nd)\} \le 1$ for some parameter $A \le O(1)$. Then we can show $\epsilon_\ell = 2\phi_0\eta nd(1 + dH\eta)^\ell = 2\phi_0\eta nde^{\ell dH\eta} \le 2\phi_0\eta nde^A < 1$. Next, we can show that compute $C \le 3d\eta(\epsilon_\ell^2 + 1) \le 3d\eta \cdot 2 < 0.5$. Thus $C^L + \cdots + C + 1 \le 2$.

Note that $3H\epsilon_\ell \in (0, 0.5)$ $\delta \le 20H\epsilon_\ell \le AH \cdot \min\{1/(HLd), 1/(\phi_0 nd)\}$

The final is $2\delta \le 2AH \cdot \min\{1/(HLd), 1/(\phi_0 nd)\} \le 2A/L$.

$\square$

**Remark D.2.** *We remark that our proof can be straightforwardly generalized to the situation where the Self Attention Network also has MLP layers, similar to Section 3.2 in (Dong et al., 2021), by defining $X_\ell^{\ell_0} = f(B_\ell^{\ell_0})$ where $f$ is the MLP layer. Note that the Lipshitz property of $f$ will appear correspondingly in the final bound.*

**Remark D.3.** *Note that in our Theorem D.1, we assume that $\eta \le O(\min\{1/(HLd), 1/(\phi_0 nd)\})$. Meanwhile, the prior work (Dong et al., 2021, Corollary 2.3) similarly requires $\beta \le O(\sqrt{d}/(H\phi_0))$.*

*Recall that $\beta$ is in terms of the $\ell_1$ norm, and so may be up to a factor of $d^2$ larger than our $\eta$; their factor of $\sqrt{d}$ only modestly helps with this.*

*(Their result statement is in terms of $\|\operatorname{Res}(X)\|_\infty$ rather than $\|X\|_\infty$, but we could state ours in terms of $\operatorname{Res}(X)$ instead; we simply bound $\|\operatorname{Res}(X)\|_\infty \le \|X\|_\infty$ in the proof of our Lemma C.2.)*

### D.2 LIPSCHITZ PROPERTY

We state a simple application of Lemma B.2.

**Corollary D.4.** *Let $a, b \in \mathbb{R}^n$. Then, we can show that*

$$\|\operatorname{softmax}(a+b) - \operatorname{softmax}(a)\|_\infty \le 2(e^{\|b\|_\infty} - 1)$$

*Proof.* The proof is same as Lemma B.2. $\qquad\square$

**Lemma D.5.** *Let $a, b \in \mathbb{R}^n$. If $\|b\|_\infty \le 1$, then we have*

$$\|\operatorname{softmax}(a+b) - \operatorname{softmax}(b)\|_\infty \le 4\|b\|_\infty$$

*Proof.* Note that for $x \in (0, 1]$, we know $e^x - 1 \le 2x$.

Thus, we know

$$\|\operatorname{softmax}(a+b) - \operatorname{softmax}(b)\|_\infty \le 2(e^{\|b\|_\infty} - 1)$$
$$\le 4\|b\|_\infty$$

where the first step follows from Corollary D.4, the second step follows from $e^x - 1 \le 2x$. $\qquad\square$

**Lemma D.6.** *If the following conditions hold*

- *Let $W_q, W_k, W_v$ denote weight matrices.*

- *Let $W = W_q W_k^\top$.*

- *Let $Y$ satisfy that $\|Y - X\|_\infty \le 2\|X\|_\infty$*

- *Let $d\|X\|_\infty \eta \le 1/4$. (This condition is verified in Lemma D.8)*

- *$K_1 := 12d\|X\|_\infty\|W\|_\infty$.*

- *$K_2 := K_1 nd\|X\|_\infty\|W_v\|_\infty + d\|W_v\|_\infty$*

*Then, we can show*

- **Part 1.**

$$\|\operatorname{softm}(X) - \operatorname{softm}(Y)\|_\infty \le K_1 \cdot \|X - Y\|_\infty$$

  *Further, $K_1 = 4$.*

- **Part 2.**

$$\|\operatorname{softmv}(X) - \operatorname{softmv}(Y)\|_\infty \le K_2 \cdot \|X - Y\|_\infty$$

*Proof.* Before going to prove each parts, we will first show

$$\|XWX^\top - YWY^\top\|_\infty \le \|XWX^\top - XWY^\top\|_\infty + \|XWY^\top - YWY^\top\|_\infty$$
$$\le \|XW\|_\infty \cdot \|X - Y\|_\infty + \|WY^\top\|_\infty \cdot \|X - Y\|_\infty$$
$$\le (\|WX\|_\infty + \|WY\|_\infty) \cdot \|X - Y\|_\infty$$
$$\le 3d\|W\|_\infty\|X\|_\infty\|X - Y\|_\infty$$
$$\le 3d\|W\|_\infty 2\|X\|_\infty^2$$
$$\le 6d^2\eta^2\|X\|_\infty^2$$

$$\le 1$$

the first step follows triangle inequality, the second step follows from Fact 3.3, the third step follows from Fact 3.3, the sixth step follows from $\|W\|_\infty \le d\eta^2$, the last step follows from assumption in the Lemma statement.

**Proof of Part 1.** We can show

$$\| \operatorname{softm}(X) - \operatorname{softm}(Y)\|_\infty = \| \operatorname{softmax}(XWX^\top) - \operatorname{softmax}(YWY^\top)\|_\infty$$
$$\le 4\|XWX^\top - YWY^\top\|_\infty$$
$$\le 12d\|W\|_\infty\|X\|_\infty\|X - Y\|_\infty$$

where the second step follows from Lemma D.5 with $\|XWX^\top - YWY^\top\|_\infty \le 1$.

**Proof of Part 2.** We can show

$$\| \operatorname{softmv}(X) - \operatorname{softm}(Y)\|_\infty$$
$$= \| \operatorname{softm}(X)XW_v - \operatorname{softm}(Y)YW_v\|_\infty$$
$$\le \| \operatorname{softm}(X)XW_v - \operatorname{softm}(Y)XW_v\|_\infty + \| \operatorname{softm}(Y)XW_v - \operatorname{softm}(Y)YW_v\|_\infty$$
$$\le \| \operatorname{softm}(X) - \operatorname{softm}(Y)\|_\infty \cdot \|XW_v\|_\infty \cdot n + \|(X - Y)W_v\|_\infty$$
$$\le K_1 n\|XW_v\|_\infty\|X - Y\|_\infty + d \cdot \|W_v\|_\infty\|X - Y\|_\infty$$
$$\le K_1 nd\|X\|_\infty\|W_v\|_\infty\|X - Y\|_\infty + d \cdot \|W_v\|_\infty\|X - Y\|_\infty$$

where the first step follows from definition, the second step follows from triangle inequality, the third step follows from Fact 3.3 and Fact E.2, and the forth step follows from Part 1 and Fact 3.3, and the last step follows from Fact 3.3.

$\square$

### D.3 INSTANTIATING AN INSTANCE FOR EACH LAYER LIPSCHITIZ PROPERTY

**Lemma D.7.** *If the following conditions hold*

- *Let $X_\ell$ denote $\ell$-th layer output*

- *Let $\|W_q\|_\infty, \|W_k\|_\infty, \|W_v\|_\infty \le \eta$*

- *Let $Y$ satisfy that $\|Y - X_\ell\|_\infty \le 2\|X_\ell\|_\infty$*

- *$\epsilon_\ell := 2\eta\phi_0(1 + dH\eta)^\ell nd$.*

*Then, we can show*

- *$\| \operatorname{softmv}(X_\ell) - \operatorname{softmv}(Y)\|_\infty \le 3d\eta(\epsilon_\ell^2 + 1)$*

*Proof.* We can show

$$\| \operatorname{softmv}(X_\ell) - \operatorname{softmv}(Y)\|_\infty \le K_2 \cdot \|X_\ell - Y\|_\infty$$

We just need to upper bound $K_2$

$$K_2 = K_1 nd\|X_\ell\|_\infty\|W_v\|_\infty + d\|W_v\|_\infty$$
$$\le 12nd^2\|X_\ell\|_\infty^2\|W\|_\infty\|W_v\|_\infty + d\|W_v\|_\infty$$
$$\le 12nd^2\|X_\ell\|_\infty^2\eta^3 + d\eta$$
$$\le 12nd^2(\phi_0 \cdot (1 + dH\eta))^{2\ell}\eta^3 + d\eta$$
$$= 3d\eta(\epsilon_\ell^2 + 1)$$

where the first step follows from the definition of $K_2$, the forth step follows from Lemma D.8, and the fifth step follows from the definition of $\epsilon_\ell$. $\square$

### D.4   EACH LAYER NORM IS NOT INCREASING MUCH

**Lemma D.8.** *If the following conditions hold*

- *Let $X_0$ denote the input of first layer of neural network, and satisfy $\|X_0\|_\infty \le \phi_0$*

- *For $\ell \in [L]$, we use $X_\ell$ to denote the $\ell$-th layer output*

- *Let $\|W_v\|_\infty \le \eta$*

*Then, we can show*

- *Part 1. For any $\ell$, $\|X_{\ell+1}\|_\infty \le \|X_\ell\|_\infty \cdot (1 + dH\eta)$*

- *Part 2. For any $\ell$, $\|X_\ell\|_\infty \le \phi_0 \cdot (1 + dH\eta)^\ell$*

- *Part 3. For any $\ell$, $\|X_\ell\|_\infty d\eta \le 1/4$*

*Proof.* **Proof of Part 1.**

For any $\ell$, we have

$$
\begin{aligned}
\|X_{\ell+1}\|_\infty = \|X_\ell + \sum_{i=1}^{H} \mathsf{softmv}_i(X_\ell)\|_\infty \\
\le \|X_\ell\|_\infty + H \cdot \|\mathsf{softmv}_i(X_\ell)\|_\infty \\
= \|X_\ell\|_\infty + H \cdot \|\mathsf{softm}_i(X_\ell) X_\ell W_{v,i}\|_\infty \\
\le \|X_\ell\|_\infty + H \cdot \|X_\ell W_{v,i}\|_\infty \\
\le \|X_\ell\|_\infty + H \cdot d \cdot \|X_\ell\|_\infty \|W_{v,i}\|_\infty \\
\le \|X_\ell\|_\infty (1 + dH\eta)
\end{aligned}
$$

where the first step follows from definition of $X_1$, the second step follows from triangle inequality, the third step follows from definition of softmv, the forth step follows from Fact E.2, and the fifth step follows from Fact 3.3, and last step follows $\|W_v\|_\infty \le \eta$.

**Proof of Part 2.**

We can show

$$
\begin{aligned}
\|X_\ell\|_\infty \le \|X_{\ell-1}\|_\infty (1 + dH\eta) \\
\le \cdots \\
\le \|X_0\|_\infty (1 + dH\eta)^\ell \\
\le \phi_0 \cdot (1 + dH\eta)^\ell
\end{aligned}
$$

where the first step follows from Part 1, the third step follows from recursively applying Part 1, and the last step follows from $\|X_0\|_\infty \le \phi_0$.

**Part 3.**

We can show

$$
\|X_\ell\|_\infty d\eta \le \eta d\phi_0 (1 + dH\eta)^\ell \le \epsilon_\ell \le 1/4
$$

where the second step follows from definition of $\epsilon_\ell$, and last step follows from guarantee of $\epsilon_\ell$.

Therefore, we complete the proof. $\square$

## E   PROOF OF LEMMA 4.1

**Lemma E.1** (Restatement of Lemma 4.1). *Let $\mathsf{Res}()$ be defined as Definition 3.4. If $\|A - B\|_\infty \le \epsilon$, then $\|\mathsf{Res}(A) - \mathsf{Res}(B)\|_\infty \le 2\epsilon$.*

*Proof.* Let $\mathbf{1}$ denote a column vector where all the entries are ones.

$\text{Res}(Z)$ acts columnwise: if $z^{(j)}$ is column $j$ of $Z$, then the $j$-th column of $\text{Res}(Z)$ is

$$\text{res}(z^{(j)}) := z^{(j)} - t(z^{(j)})\mathbf{1}$$

where $t(z^{(j)})$ minimizes $\|z^{(j)} - t \cdot \mathbf{1}\|_\infty$.

So it suffices to prove that for vectors $x, y \in \mathbb{R}^n$ we have

$$\|\text{res}(x) - \text{res}(y)\|_\infty \leq 2 \cdot \|x - y\|_\infty.$$

For any vector $z \in \mathbb{R}^n$, let $M(z) = \max_i z_i$, $m(z) = \min_i z_i$.

The scalar $t$ minimizing $\|z - t\mathbf{1}\|_\infty = \max_i |z_i - t|$ is

$$t(z) := \frac{M(z) + m(z)}{2}$$

so $\text{res}(z) = z - t(z)\mathbf{1}$.

Let $\delta := \|x - y\|_\infty$. Then

$$|M(x) - M(y)| \leq \delta, |m(x) - m(y)| \leq \delta$$

so

$$|t(x) - t(y)| = |\frac{1}{2}(M(x) + m(x) - M(y) - m(y))|$$
$$\leq \frac{1}{2}(\delta + \delta)$$
$$= \delta.$$

Now

$$\text{res}(x) - \text{res}(y) = (x - y) + (t(x) - t(y))\mathbf{1},$$

so for each $i$,

$$|\text{res}(x)_i - \text{res}(y)_i| \leq |x_i - y_i| + |t(x) - t(y)| \leq \delta + \delta = 2\delta$$

Hence

$$\|\text{res}(x) - \text{res}(y)\|_\infty \leq 2 \cdot \|x - y\|_\infty.$$

Applying this to each column of $A, B$ shows

$$\|\text{Res}(A) - \text{Res}(B)\|_\infty \leq 2 \cdot \|A - B\|_\infty.$$

Thus, we complete the proof. $\square$

**Fact E.2.** *Given $A \in \mathbb{R}^{n \times n}$ and $B, C \in \mathbb{R}^{n \times d}$, we have $\|\text{softmax}(A)(B - C)\|_\infty \leq \|B - C\|_\infty$*

*Proof.* We just need to prove for the case: one row of $A$ and one column of $B, C$.

$$|\langle \text{softmax}(a), b - c \rangle| \leq \sum_{i=1}^n \text{softmax}(a)_i |b_i - c_i| \leq \sum_{i=1}^n \text{softmax}(a)_i \|b - c\|_\infty = \|b - c\|_\infty$$

Thus, we complete the proof. $\square$

## F    BROADER IMPACT

Our results offer new theoretical insights into the expressiveness of attention mechanisms in transformers. These findings may guide the future design of large language models toward more expressive architectures. We do not foresee any potential negative societal impacts from this work.

## G    LLM USAGE DISCLOSURE

LLMs were used only to polish language, such as grammar and wording. These models did not contribute to idea creation or writing, and the authors take full responsibility for this paper's content.

