# OpenReview forum: "Only Large Weights (And Not Skip Connections) Can Prevent the Perils of Rank Collapse"
_ICLR.cc/2026/Conference — Submitted to ICLR 2026_

### Official Review · Reviewer_cfu9 · 2025-10-15

**Soundness:** 2
**Presentation:** 3
**Contribution:** 2
**Rating:** 4
**Confidence:** 3

**Summary:**

This paper introduces the concept of "layer collapse" and proves that transformers with small weights can be approximated by a single-layer network, even when skip connections are present. The work challenges the conventional interpretation that skip connections alone prevent representational weaknesses in attention-based architectures.

**Strengths:**

- The theoretical contribution extends prior work on rank collapse by showing that small weights, rather than lack of skip connections, are the fundamental limitation. The introduction of layer collapse as a concept provides a new lens for understanding transformer expressivity, with the main theorem (Theorem 1.2) offering a formal bound relating weight magnitude (l∞ norm) to approximation error.
  - The paper offers practical insights for the design of transformers, particularly regarding weight quantization and pruning strategies. By connecting theoretical results about small weights to the extensive literature on fast attention algorithms, the work bridges complexity theory and practical implementation concerns.

**Weaknesses:**

- The theoretical analysis is restricted to relatively simple settings with bounded l∞ norms on weight matrices, while the main theorem assumes specific relationships between network depth, number of heads, and weight bounds. The paper would benefit from empirical validation showing when layer collapse actually occurs in practical transformer models trained on real tasks, rather than relying primarily on the theoretical framework. For instance, experiments measuring the approximation quality of single-layer networks for pre-trained language models at different quantization levels would strengthen the claims.
  - The connection between l∞ weight bounds and the low-rank approximation assumptions used in fast attention algorithms needs clearer exposition. While the paper cites work showing that small weights enable subquadratic algorithms, it does not sufficiently explain why the specific l∞ bound (rather than other norms like l1 or Frobenius) is the relevant constraint for both computational efficiency and layer collapse. Additional analysis or experiments comparing different norm constraints would clarify this relationship.
  - The paper's discussion of skip connections may be somewhat misleading. While the title emphasizes that "only large weights and not skip connections" prevent collapse, the actual theoretical result (Theorem 1.2 and its proof) applies to networks with or without skip connections under small weight constraints. The paper could more carefully distinguish between: (1) networks without skip connections that have exponentially fast rank collapse (Dong et al.), (2) networks with skip connections but small weights that have layer collapse (this work), and (3) networks with skip connections and large weights (unclear from current work). The relationship between these three regimes deserves more detailed discussion.
  - The experimental validation focuses primarily on correlation studies between weight norms and Hessian eigenvalues in small networks, but does not directly validate the layer collapse phenomenon itself. Missing are experiments that: (1) explicitly construct the approximating single-layer network S' for a given multi-layer network S with small weights, (2) measure the actual approximation error ||S(X) - S'(X)||∞ on realistic inputs, and (3) demonstrate at what threshold of weight magnitude η the layer collapse becomes practically relevant. The GPT2 experiments in the appendix appear to be from a different paper (about Hessian spectra) and may have been included by mistake, as they do not directly address layer collapse. I will reconsider my score in the rebuttal.

**Questions:**

see weaknesses

---

> ### Author Response · Authors · 2025-11-17
> **Reply to Reviewer cfu9**
>
> Thank you for your thoughtful review. To address the weaknesses:
>
> * Our main result mathematically proves that layer collapse occurs in networks with small weights. This is stronger than any experiment can demonstrate, since it provably applies to all networks with small weights, including any transformer models trained on real tasks that one can imagine. We feel that asking for experiments is missing the point here; we are giving a theoretical explanation for a fundamental question of which aspects of self-attention lead to their expressive power, showing that large weights, rather than skip connections, are important. In particular, layer collapse is a strong representational strength weakness, and we would not expect transformers used in practice to experience it (since otherwise they would be too weak to use in practice!) just as rank collapse from prior work is also not observed in practical transformers. We emphasize that our result applies to a strict superset of the networks that (Dong et al.) applies to, including networks with skip connections.
> * An l∞ bound on the weight matrices is the best one could hope for in our type of theorem. For instance, the l∞ norm is always smaller than the l1 or Frobenius norm of the matrix. Our theorem thus classifies the largest possible set of matrices as having “small entries”, and using one of these other norms would weaken the result. The prior work on algorithms that you mention also uses the l∞ norm of the matrices.
> * The main point is that the prior work applied only to networks with small weights and no skip connections, whereas our result applies to networks with small weights even if they do have skip connections. Because of this, our result is strictly stronger. Another way of saying this: the category (1) of networks you suggest must also have small weights (since (Dong et al.) only proves rank collapse for networks with small weights and no skip connections), so your category (1) is a strict subset of your category (2). (Then indeed, for your category (3), networks with large weights almost always do not have rank collapse, whether or not there are skip connections, and the main takeaway of our paper is that one should use large weights to be in category (3).)
> * You write that “The experimental validation focuses primarily on correlation studies between weight norms and Hessian eigenvalues in small networks, but does not directly validate the layer collapse phenomenon itself” and later that “The GPT2 experiments in the appendix appear to be from a different paper (about Hessian spectra) and may have been included by mistake, as they do not directly address layer collapse.” We’re not sure what this is referring to; there are no experimental validations, or references to Hessian eigenvalues or GPT2 in our paper. Perhaps, indeed, you were looking at a different paper?

---

### Official Review · Reviewer_G6Fb · 2025-10-30

**Soundness:** 1
**Presentation:** 1
**Contribution:** 2
**Rating:** 0
**Confidence:** 4

**Summary:**

This theoretical paper aims to prove that the assumption of many previous papers approximating attention with sub-quadratic algorithms -- that the weights of the model are "small" -- is not realistic. Limiting the weights like this would (according to the main Theoreom in this paper) lead to rank collapse, which implies that attention with "small" weights is less expressible than attention without such constraints.

**Strengths:**

The main motivation of this paper is strong and -- if the main Theorem was proven properly -- this paper would make a great contribution to the field. I believe that studying the "layer collapse" instead of the "rank collapse" (as defined by the authors) makes a lot of sense and could lead to interesting discoveries in the future.

**Weaknesses:**

The paper is poorly structured and full of typos and logical mistakes.

Right the first Lemma 4.1 is incorrect and the proof has a mistake in the first inequality. The Lemma states that if $||A-B||\_{\\infty} \\leq \epsilon$, then $||Res(A) - Res(B)||\_{\\infty} \\leq \epsilon$. A counterexample to this is setting $A = \\begin{pmatrix} 2 & 0 & -2 \\end{pmatrix}^T$ and $B = \\begin{pmatrix} 1 & 1 & -3 \\end{pmatrix}^T$; then $Res(A) = A - 0 = A$ and $Res(B) = B - (-1) = \\begin{pmatrix} 2 & 2 & -2 \\end{pmatrix}^T$. This means that $||A-B||\_{\\infty} = ||\begin{pmatrix} 1 & -1 & 1 \\end{pmatrix}^T||\_{\\infty} \\leq 1$, but $||Res(A) - Res(B)||\_{\\infty} =  ||\begin{pmatrix} 0 & -2 & 0 \\end{pmatrix}^T||\_{\\infty} \\not\\leq 1$, which is a contradiction to the Lemma. Furthermore, the proof in the paper implies that $||Res(A) - Res(B)||\_{\\infty} \\leq ||A-B||\_{\\infty}$, which is incorrect for the same reason.

This Lemma 4.1 is then used to prove the main Lemma 5.2, it is applied on line 442 -- making the proof of the main Theorem incorrect. One inequality on this line claims $||Res(B - R\_A) - Res(B)||\_{\\infty} \\leq ||(B - R\_A) - B||\_{\\infty} = ||R\_A||\_{\\infty}$, but this is not always true, as shown above.

The mistakes appear in many other places, most importantly in the central Lemmas 5.2 and C.1. These Lemmas confuse the definition of $\\text{softm}$ (regular parameter-less softmax operator as defined in 3.1) with the definition of $\\text{SAtt}$ (parametrized self-attention operator defined in 3.6) -- the formulation of Lemma 5.2 talks about weights $W\_k, W\_q$ and $W\_v$, but there are clearly no weights involved in $\\text{softm}$. In order to prove the main Theorem, we would need to prove that $||\\text{SAtt}(B) - \\text{SAtt}(X)||\_{\\infty} \\leq \epsilon_0$, but these Lemmas only prove that $||\\text{softm}(B) - \\text{softm}(X)||\_{\\infty} \\leq \epsilon_0$ and $||\\text{softmv}(B) - \\text{softmv}(X)||\_{\\infty} \\leq \epsilon_0$, which is not enough.

____

Apart from this, the structure and writing of this paper also lacks; starting from the abstract that contains badly formatted citations. The Related Work section spans almost two pages and contains mostly irrelevant papers.

**Questions:**

- I am open to believing that what appears as mistakes in the paper are just unfortunate typos, could you explain how can the Lemma 4.1 be true?
- Similarly, can you prove Lemma 5.2 for $\\text{SAtt}$ instead of $\\text{softm}$?

---

> ### Author Response · Authors · 2025-11-17
> **Reply to Reviewer G6Fb**
>
> Thank you for your thoughtful review. In our attempt to simplify some of the arguments and notation, especially in the first 9 pages, we sloppily missed important constants or notation. Thank you for carefully noticing this. We have uploaded a revised version in which all the issues you mention are resolved, and we believe all our proofs are now correct as written. The changes are in blue in the revision compared to the original submission.
>
> In particular:
> * Lemma 4.1 was missing a constant factor of 2, i.e., the difference in Res is never more than twice the difference in the original matrices. This indeed adds (insignificant) factors of 2 to the other locations where we use this lemma later in the paper. These have all been fixed in the updated version.
> * We have clarified the definitions of softm vs softmax vs SAtt (see definitions 3.1 and 3.6) and used the correct versions in each of the appropriate locations. We originally had softm instead of SAtt because we aimed to somewhat simplify the notation, but you’re right that the weight matrices are needed for the full meaning of the theorem statement, so we have included them in the statements and proofs. The key lemma for incorporating weight matrices in the proofs appeared already as Lemma D.6 in our submission, and we have referenced it in the appropriate places where it is used.
>
> Related Works: Our goal of this section is to demonstrate that the topics we’re studying in this paper are broadly and popularly studied in the literature, both to emphasize the reach of our results, and to clarify that we are not the first to come up with some of these concepts. For example, for diffusion models, the point we are aiming to make is that (1) most diffusion models use transformer backbones, meaning our results apply to them, and (2) a line of prior work has studied theoretical guarantees on efficiently approximating diffusion models, which our work fits in to. We feel discussing these is important to explain the context of our results, but are happy to shorten these sections.

---

> ### Comment · Reviewer_G6Fb · 2025-11-18
>
> Dear authors, thank you for swiftly responding and updating the paper. Before I have time to write a proper response, I need to say that there is most likely still a major mistake in your proof. The issue is with Fact 3.2, which holds for `softmax` function, but not for the newly introduced `softm` function. As a counterexample, we can set the weight matrices to be identity matrices and have $X=(-1\quad1)^T$ and $\alpha=1$. You use Fact 3.2 seven times in the proofs for the main theorem, but always incorrectly for `softm` instead of `softmax`.

---

> > ### Author Response · Authors · 2025-11-19
> >
> > Thank you for reading carefully and pointing that out. In fact, we believe this is a minor issue which also appears in the proofs in the original (Dong et al.) paper! (It arises when generalizing to multiple layers, where the relevant details aren't given in that paper.) At any rate, this is not hard to fix: to translate from softmax to softm, use that || X - Y|| is small in order to bound that ||X W X^\top - Y W Y^\top|| is also small. We have fixed this in the updated pdf, with the changes again in blue.

---

### Official Review · Reviewer_CG8P · 2025-10-31

**Soundness:** 2
**Presentation:** 3
**Contribution:** 2
**Rating:** 4
**Confidence:** 2

**Summary:**

The paper proves that self-attention networks with small weights lose depth expressivity even when skip connections are used. It shows that if weight magnitudes are bounded, an L-layer model effectively behaves like a single-layer one. This challenges the common belief that skip connections alone prevent rank collapse.

**Strengths:**

- Provides a clear and rigorous theoretical result on the limits of skip connections under small-weight conditions.
- Elegant and well-structured proofs using softmax perturbation and layer-removal arguments.
- Corrects a major misconception in prior work (Dong et al., 2021).

**Weaknesses:**

- No empirical validation or quantitative mapping of η to realistic settings.
- Purely theoretical, lacks visualizations or verification experiments like Dong et al. (2021).
While Dong et al. provided practical insight (“skip connections are necessary”), this paper mainly corrects a misconception without offering clear actionable guidance for model design or training.
- The paper also contains unexplained square symbols in several places.

**Questions:**

I’m curious what practical contribution this paper can offer to the community beyond correcting a misconception.
As noted in the paper, many real-world LLMs use approximation or quantization techniques that enforce weight bounds, which can indeed be risky. However, such risks are typically addressed through validation accuracy and calibration during deployment.
Given that, I wonder what unique or practical value this work adds to the community beyond highlighting this theoretical limitation.

---

> ### Author Response · Authors · 2025-11-17
> **Reply to Reviewer CG8P**
>
> Thank you for your thoughtful review. To address the weaknesses:
>
> * Our main result mathematically proves that layer collapse occurs in networks with small weights. This is stronger than any experiment can demonstrate, since it provably applies to all networks with small weights, including any transformer models trained on real tasks that one can imagine. We feel that asking for experiments is missing the point here; we are giving a theoretical explanation for a fundamental question of which aspects of self-attention lead to their expressive power, showing that large weights, rather than skip connections, are important. In particular, layer collapse is a strong representational strength weakness, and we would not expect transformers used in practice to experience it (since otherwise they would be too weak to use in practice!) just as rank collapse from prior work is also not observed in practical transformers. We emphasize that our result applies to a strict superset of the networks that (Dong et al.) applies to, including networks with skip connections.
> * A main message of our paper is that the “practical insight” that you say (Dong et al.) provides (that “skip connections are necessary”) is actually wrong! Correcting an incorrect practical insight from prior work may already be seen as a practical insight. Moreover, we show that small weights lead to layer collapse, and so another “practical insight” is that people designing or training models should be sure to allow sufficiently large weights in their models.
> But for what it’s worth, we also strongly disagree that “purely theoretical” is a weakness. Developing a theory of which aspects of self attention networks lead to representation strength is an incredibly important challenge which we work to address here. Moreover, the ICLR call for papers specifically welcomes theoretical papers.
> * We believe the “square symbols” you are referring to are the standard end-of-proof symbols used at the ends of proof environments in most mathematical papers including the ICLR latex style. See https://en.wikipedia.org/wiki/Tombstone_(typography)
>
> Question:
> * We mentioned some practical takeaways in response to the second weakness above. But we primarily view our result as having a strong theoretical contribution, which we hope will lead to more practical contributions in the future. One of the most basic challenges in the theory of LLMs is understanding which components lead to the surprisingly robust expressive power of transformer and other similar models. Very little is known about this, and our result highlights one part (large weights) which is provably important, while also pointing out one part (skip connections) that are not as important as previously believed. Understanding which components lead to expressive power is not only an interesting question in its own right, but also can help direct future work on defining new or better models (or perhaps even simpler models, by removing unimportant components). For one comparison, the prior work of (Dong et al.) has been cited in hundreds of follow-up works which it helped to motivate.

---

### Official Review · Reviewer_rupd · 2025-10-31

**Soundness:** 1
**Presentation:** 1
**Contribution:** 1
**Rating:** 0
**Confidence:** 5

**Summary:**

This paper tries to show that multi-layer attention models with only small weights can be collapsed with a small error into a single-layer attention model. The authors posit that only if models contain large weights can they be safe from layer collapse (a variant of rank collapse where all layers can be collapsed into a single collapse).

**Strengths:**

The idea of this paper is interesting and could, if shown, affect the way models are pre-trained since it posits that using only small weights can lead to layer collapse.

**Weaknesses:**

Minor: The related works are quite confusing; the three paragraphs contain mostly related works (although they sometimes include works on topics that do not relate to the paper, such as privacy), but the next three paragraphs talk about works that are not related to the paper in anyway (such as diffusion or regression models when the paper is about the importance of large weights to avoid layer collapse). This adds half a page of irrelevant text to the paper.

Minor/Major: Lemma 4.1 and Lemma 4.2 seem to be unused and irrelevant to the paper; it is confusing as to why they are in the main paper rather than the appendix.

Major: Most of the mathematics is wrong; the proof of Lemma 4.1 is incorrect (if you take $A = [-2,4,-2], B=[-2, 2, 2]$ then $\Vert{A -B}\Vert= 4$ and $\Vert\textrm{Res}(A) - \textrm{Res}(B)\Vert=5$). The important steps tend to be under-explained, while the trivial ones are detailed. Both Lemma 5.1 and 5.2 (which are the main lemmas of the paper) contain mistakes in the proof, which render the theorem of the paper unproven. Lemma 5.1 has multiple typos, missing addition signs, and incorrect order of matrix multiplication in the proof. In lines 404-406, you have the D matrix appear out of thin air without further explanation (or it coming from the previous part of the proof). You assume that you can replace $(e^D - 1)$ by $(e^{\theta} - 1)$ when taking the infinite norm, but this however not the case since if $\theta < 0.69$ then the infinite norm of the matrix is 1. For Lemma 5.2, in line 442-443, you state that $\Vert \textrm{Res}(B - R_A) - \textrm{Res}(B)\Vert \leq \Vert R_A \Vert$. This is false; the inequality should be in the other direction by definition of the Res function. In addition, you use the function softmax for Lemma 5.2 instead of Self Attention, which does not allow you to do Equation (9).

**Questions:**

- Why were Lemmas 4.1 and 4.2 included in the paper? Are they relevant to any part of the further proofs (I could not find any references to them)?

---

> ### Author Response · Authors · 2025-11-17
> **Reply to Reviewer rupd**
>
> Thank you for your thoughtful review.
>
> Related Works: Our goal of this section is to demonstrate that the topics we’re studying in this paper are broadly and popularly studied in the literature, both to emphasize the reach of our results, and to clarify that we are not the first to come up with some of these concepts. For example, for diffusion models, the point we are aiming to make is that (1) most diffusion models use transformer backbones, meaning our results apply to them, and (2) a line of prior work has studied theoretical guarantees on efficiently approximating diffusion models, which our work fits in to. We feel discussing these is important to explain the context of our results, but are happy to shorten these sections.
>
> Major comments: In our attempt to simplify some of the arguments and notation, especially in the first 9 pages, we sloppily missed important constants or notation. Thank you for carefully noticing this. We have uploaded a revised version in which all the issues you mention are resolved, and we believe all our proofs are now correct as written. The changes are in blue in the revision compared to the original submission.
>
> In particular:
> *  Lemma 4.1 was missing a constant factor of 2, i.e., the difference in Res is never more than twice the difference in the original matrices. This indeed adds (insignificant) factors of 2 to the other locations where we use this lemma later in the paper. These have all been fixed in the updated version.
> * We have elaborated on a few more proof details, but if there are important steps which you still think are under-explained, it would help us if you could tell us which ones so that we can address them.
> * Lemmas 5.1 and 5.2 should be fully correct. We apologize about the typos, but we believe the proofs are fully correct.
>
> * The matrix D is a diagonal matrix with all nonnegative entries which are bounded above be theta, so e^theta - 1 does give an upper bound on the infinite norm of e^D - 1. We have elaborated on this in the proof.
>
> * Lemma 5.2: The inequality you mention was using Lemma 4.1; we have updated it to appropriately have the factor of 2.
>
> * We have clarified the definitions of softm vs softmax vs SAtt (see definitions 3.1 and 3.6) and used the correct versions in each of the appropriate locations, such as lemma 5.2 where you pointed this out.
>
> Question:
> * Lemmas 4.1 and 4.2 are used in many places throughout the paper where we need to bound differences of Res. We wanted to provide these early on so that the reader would understand what we were using whenever such a bound arose. For example, Lemma 4.1 is being used in the proof of Lemma 5.2 (see the updated version, line 443 in page 9, page 25) and Lemma 4.2 is being used in the proof of Lemma B.1 (see updated version page 22)

---

### Meta-Review · Area_Chair_iCKo · 2026-01-05

**Summary:**

The authors study the phenomenon of layer and rank collapse in deep transformers, and makes the claim that **only larger weights** can prevent this issue. This is quite a strong claim as it is known that stable infinite depth limits of transformers have been found both with and without skip connections, see e.g. https://arxiv.org/abs/2306.17759 and https://arxiv.org/abs/2405.15712. In particular, it doesn't seem like "large" or "small" weights played a central role, but rather an appropriate scaling of depth or non-linearity is the key.

Putting that aside for now, the reviewers nonetheless have been quite critical of this work for several important issues:
 - Mathematical correctness.
 - Sloppy writing in key definitions and explanations.
 - Poor presentation quality.

There are other minor issues but I will just focus on these three for now.

From what I can tell, there was a critical error in Lemma 4.1 found by multiple reviewers, which the authors claimed is fixed. However, an additional critical issue was caught for Fact 3.2. These are just the main issues found by the reviewers.

A more careful read can find many more additional issues:
 - Lemma B.4 is not proved as it is stated, and likely false, due to definition mismatch and missing factor of $d$
 - Lemma D.6 $K_1 = 4$ is is unjustified
 - Lemma D.8 $\epsilon_\ell \leq \frac{1}{4}$ is not justified
 - Inconsistent use of $\epsilon_\ell$ definitions in different sections

Overall, this confirms the reviewers comments on this submission involving too much sloppy writing in the mathematical content. All of this is to say, this manuscript is **not ready for publication** based on a long list of careless mistakes. I hope the authors take more time to refine their work before resubmitting, as this is consuming a lot of reviewing efforts from the conference reviewers and ACs.

**Reviewer Concerns:**

Addressed:
- Error in Lemma 4.1
- Confusion Between softm and SAtt
- Inequality Direction in Lemma 5.2

Outstanding:
- Validity of the Main Proof (Fact 3.2 / softm, and additional issues raised above)
- Lack of Empirical Validation
- Unexplained square symbols
- Skip-connection framing / distinguishing regimes

**Reviewer Scores:**

rupd - 0
G6Fb - 0
CG8P - 4
cfu9 - 4

I don't believe any of the reviewers are likely to raise their score.

---

### Decision · Program_Chairs · 2026-01-26

Reject